# PointTruss: K-Truss for Point Cloud Registration

**Yue Wu**[1,2]     **Jun Jiang**[1,2]     **Yongzhe Yuan**[1,2]     **Maoguo Gong**[1,3,5*]
**Qiguang Miao**[1,2]     **Hao Li**[1,3]     **Mingyang Zhang**[1,3]     **Wenping Ma**[4]

[1]MoE Key Lab of Collaborative Intelligence Systems, Xidian University
[2]School of Computer Science and Technology, Xidian University
[3]School of Electronic Engineering, Xidian University
[4]School of Artificial Intelligence, Xidian University
[5]Academy of AI, College of Mathematics Science, Inner Mongolia Normal University
{ywu, qgmiao, haoli, myzhang, wpma}@xidian.edu.cn, gong@ieee.org,
xiaolongfan@outlook.com, {junj, yyz}@stu.xidian.edu.cn

## Abstract

Point cloud registration is a fundamental task in 3D computer vision. Recent advances have shown that graph-based methods are effective for outlier rejection in this context. However, existing clique-based methods impose overly strict constraints and are NP-hard, making it difficult to achieve both robustness and efficiency. While the k-core reduces computational complexity, which only considers node degree and ignores higher-order topological structures such as triangles, limiting its effectiveness in complex scenarios. To overcome these limitations, we introduce the $k$-truss from graph theory into point cloud registration, leveraging triangle support as a constraint for inlier selection. We further propose a consensus voting-based low-scale sampling strategy to efficiently extract the structural skeleton of the point cloud prior to $k$-truss decomposition. Additionally, we design a spatial distribution score that balances coverage and uniformity of inliers, preventing selections that concentrate on sparse local clusters. Extensive experiments on KITTI, 3DMatch, and 3DLoMatch demonstrate that our method consistently outperforms both traditional and learning-based approaches in various indoor and outdoor scenarios, achieving state-of-the-art results.

## 1  Introduction

Point cloud registration is a fundamental problem in 3D computer vision [43, 29], remote sensing [19, 12], 3D reconstruction [6, 11], and autonomous driving [50]. Its primary goal is to estimate the optimal rigid transformation matrix that precisely aligns two point clouds. Accurate 3D correspondences form the foundation of point cloud registration. High-quality correspondences enable the correct computation of rotation and translation, and their quality directly affects the final registration accuracy.

Recent works [48, 1] demonstrate the effectiveness of graph theory for correspondences selection in point cloud registration. In a graph, vertices represent matched point pairs, and edges encode geometric compatibility, which are added when two correspondences meet a preset threshold [30]. Based on this representation, several graph algorithms have been developed to select reliable correspondences. The classical maximal clique approach [49, 37] seeks reliable matches by finding fully connected subgraphs, ensuring strong geometric consistency but facing two major challenges. The strict connectivity is often unachievable due to noise or occlusion, resulting in the loss of correct matches. Maximal clique search is NP-hard [13], making it impractical for large-scale data. The k-core method [35, 32] relaxes connec-

---

*Corresponding author.

tivity requirements and reduces computation, but its weak constraints compromise correspondences reliability, which highlights the difficulty in balancing structural robustness and efficiency.

To address these limitations, we introduce the $k$-truss concept from community detection [47, 46, 26] into point cloud registration. A $k$-truss [9] requires each edge to participate in at least $k$-2 triangles, leveraging the rigidity and invariance of triangles. Triangles derive their strength from their stable structure, serving as the simplest rigid planar formation and preserving their geometric properties regardless of rotation or translation. In real-world data, inlier correspondences naturally form triangle-rich clusters, as shown in Fig. 1, making $k$-truss decomposition effective for preserving reliable matches while filtering outliers. Our method first uses consensus voting for low-scale sampling, then constructs a compatibility graph and applies $k$-truss decomposition. For each resulting subgraph, we estimate the transformation via weighted SVD [7] and rank candidates using a spatial distribution score, selecting the transformation with the highest score as the final result.

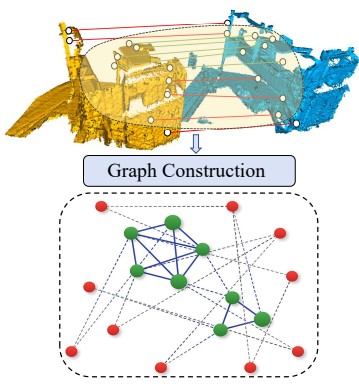

Figure 1: **Triangular structure as inlier indicator.** Inlier correspondences (green) form triangle-rich structures, while outliers (red) lack triangular support.

To our knowledge, this is the first work to introduce $k$-truss into point cloud registration, establishing a robust correspondences selection framework based on triangle constraints. Inspired by $k$-truss, we propose a heuristic method that leverages triangle-based truss structures to robustly filter and refine correspondences. Experimental results show that our method achieves excellent performance under high noise and outlier ratios. Meanwhile, it maintains polynomial time complexity and demonstrates high computational efficiency among graph-based methods. Extensive evaluations on standard datasets further demonstrate that our method significantly outperforms existing state-of-the-art approaches in registration accuracy. Our main contributions are summarized as follows:

- We propose a novel correspondence selection method called PointTruss. It uses triangle support constraints to effectively filter out mismatches related to isolated and low-support edges.
- We develop an integrated pipeline. It applies consensus-voting low-scale sampling to extract the structural skeleton, $k$-truss decomposition to preserve triangle-supported inliers, and spatial distribution scoring to favor broad, uniform coverage. Each component is modular and can be used independently.
- Extensive experiments on KITTI, 3DMatch, and 3DLoMatch show that PointTruss consistently outperforms both traditional and learning-based methods across diverse indoor and outdoor scenarios. It achieves state-of-the-art accuracy and efficiency with polynomial-time complexity and strong robustness to noise and outliers.

## 2   Related Work

**Traditional Point Cloud Registration.** RANSAC and its variants [14, 2] iteratively sample from the initial correspondence set to find the largest consensus set. Early handcrafted feature descriptors, such as FPFH [31], extract local features by encoding geometric histograms. FGR [51] estimates the optimal transformation using robust estimators like the Geman-McClure loss. Branch-and-bound (BnB) based optimization methods, such as GORE [3] and its variant QGORE [25], perform global search in the parameter space to obtain the best transformation. Voting-based method [40] select reliable correspondences through a scoring mechanism. Some works tackle registration with high outlier rates and non-convex objectives via robust and global search, mitigating local minima and improving convergence [28, 34]. However, these methods often suffer from low computational efficiency and limited accuracy under high outlier ratios.

**Learning-based Point Cloud Registration.** Current learning-based point cloud registration approaches can be categorized as follows. The first category focuses on detecting reliable keypoints or extracting more discriminative features [29, 41]. For example, FCGF [8] uses a fully convolutional network to extract point cloud features in a single pass, without separate keypoint detection. Another category aims to distinguish inliers from outliers. PointDSC [1] removes outliers using pairwise spatial compatibility supervision, while VBReg [20] introduces variational non-local networks for

outlier rejection. There are also end-to-end approaches [42], such as Deep Global Registration (DGR) [7], which employs sparse convolution and point-wise MLPs to classify correspondences. Although these methods perform well in specific scenarios [44], they generally require large amounts of training data and have limited generalization ability. In contrast, training-free graph-based registration methods often exhibit better robustness and can be integrated as auxiliary modules in deep learning frameworks to further improve overall performance.

**Graph-based Point Cloud Registration.** Graph-based algorithms [30] typically construct a compatibility graph by evaluating the pairwise compatibility of correspondences, which enables efficient removal of a large number of outliers. For example, the TEASER [39] employs maximal clique theory to decouple scale, rotation, and translation estimation. ROBIN [32] uses the maximal k-core theory for outlier pruning. SUCOFT [35] introduces the concept of k-supercore to improve outlier rejection effectiveness. SC$^2$-PCR [5] imposes stricter constraints on correspondences by introducing a second-order spatial compatibility metric. MAC [48] first proposes a maximal clique-based method to mine richer local consistency information, while FastMAC [49] accelerates computation by applying random spectral sampling on the correspondence graph. These methods demonstrate that mining key information in the compatibility graph is crucial for improving the robustness and accuracy of point cloud registration.

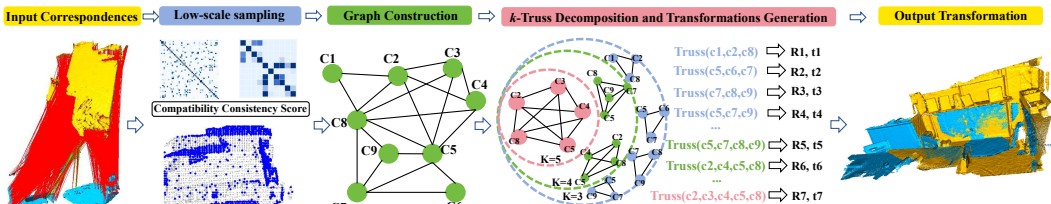

Figure 2: **Pipeline of our method.** 1. Starting from input correspondences, perform low-scale sampling to reduce redundancy. 2. Constructing a correspondence graph and apply $k$-truss decomposition to identify subgraphs with varying levels of triangle support (i.e., different $k$ values). 3. Each $k$-truss subgraph represents a set of correspondences with a specific degree of structural consistency. 4. Based on these subgraphs, generate multiple transformation hypotheses and select the optimal transformation using the spatial distribution score.

## 3 Methods

### 3.1 Problem Formulation

For two point clouds, where the source point cloud is defined as $\mathbf{P} = \{\mathbf{p}_i \in \mathbb{R}^3 \mid i = 1, \dots, N\}$ and the target point cloud as $\mathbf{Q} = \{\mathbf{q}_i \in \mathbb{R}^3 \mid i = 1, \dots, M\}$, the goal of point cloud registration is to estimate the rigid transformation $\mathbf{T} = \{\mathbf{R}, \mathbf{t}\}$ that aligns these two point clouds. Here, $\mathbf{R} \in SO(3)$ represents the rotation matrix, and $\mathbf{t} \in \mathbb{R}^3$ represents the translation vector. The optimization problem can be formulated as:

$$\min_{\mathbf{R}, \mathbf{t}} \sum_{(\mathbf{p}_i, \mathbf{q}_i) \in \mathcal{C}} \|\mathbf{R}\mathbf{p}_i + \mathbf{t} - \mathbf{q}_i\|_2^2, \tag{1}$$

where $\mathcal{C} = \{\mathbf{c}_i \mid i = 1, \dots, N_c\}$ is the initial correspondence set obtained through feature matching, with each correspondence $\mathbf{c}_i = (\mathbf{p}_i, \mathbf{q}_i)$.

We extract either geometric or learned local features from the point clouds, use feature matching to generate $\mathcal{C}$, and apply the $k$-truss method to extract the optimal subgraph. This subgraph is then used to estimate the six degrees of freedom (6-DoF) pose transformation between $\mathbf{P}$ and $\mathbf{Q}$. The overall pipeline is illustrated in Fig. 2, and the PointTruss is both simple and efficient.

### 3.2 Graph Construction

The graph space can more accurately capture the affinity relationships between correspondences than Euclidean space [5]. Therefore, we represent the initial correspondences as a compatibility

graph, where each node denotes a correspondence and edges connect nodes that are geometrically compatible [1, 23, 24, 30, 48].

The graph is constructed using the rigid distance constraint between correspondence pairs $(\mathbf{c}_i, \mathbf{c}_j)$, which is quantitatively measured as:

$$S_{dist}(\mathbf{c}_i, \mathbf{c}_j) = |\|\mathbf{p}_i - \mathbf{p}_j\| - \|\mathbf{q}_i - \mathbf{q}_j\|| \leq 2\tau, \tag{2}$$

where $\tau = c \cdot \sigma$ is the distance threshold, $c$ is typically set to 3.5 based on statistical confidence intervals, and $\sigma$ is the standard deviation of noise, which controls the sensitivity to distance discrepancies. This constraint ensures that the distance between point pairs remains nearly invariant under rigid transformations.

The compatibility consistency score between $\mathbf{c}_i$ and $\mathbf{c}_j$ is defined as:

$$S_{comp}(\mathbf{c}_i, \mathbf{c}_j) = \exp\left(-\frac{S_{dist}(\mathbf{c}_i, \mathbf{c}_j)^2}{2\sigma^2}\right), \tag{3}$$

If $S_{comp}(\mathbf{c}_i, \mathbf{c}_j)$ exceeds a threshold $\tau_{comp}$, an edge $\mathbf{e}_{ij}$ is formed between $\mathbf{c}_i$ and $\mathbf{c}_j$. The weight of the edge is $S_{comp}(\mathbf{c}_i, \mathbf{c}_j)$. Otherwise, $S_{comp}(\mathbf{c}_i, \mathbf{c}_j)$ is set to zero.

### 3.3 Consensus Voting-based Low-scale Sampling Strategy

To efficiently identify inlier correspondences and reduce the search space, we propose a consensus voting-based low-scale sampling strategy. This method leverages the previously defined geometric consistency metrics to identify the most reliable correspondences.

**Consensus Score Computation.** Utilizing the compatibility score $S_{comp}$ defined in Eq. (3), we compute the consensus score for each correspondence $i$ by counting the number of other correspondences that are geometrically compatible with it:

$$S_i = \sum_{j=1, j \neq i}^{N} \mathbb{I}(S_{comp}(\mathbf{c}_i, \mathbf{c}_j) > \tau_c), \tag{4}$$

where $\mathbb{I}(\cdot)$ is the indicator function and $\tau_c$ is the consistency threshold. This score represents the number of other correspondences that support correspondence $i$.

**Non-Maximum Suppression.** To avoid sampling spatially clustered correspondences, we apply non-maximum suppression [27]:

$$\text{IsLocalMax}_i = \min_{j \in \mathcal{N}} \left((S_i \geq S_j) \vee (d_{ij}^s \geq r_{\text{nms}})\right), \tag{5}$$

where $\mathcal{N}$ is the set of all correspondences, $d_{ij}^s = \|\mathbf{p}_i - \mathbf{p}_j\|_2$ is the Euclidean distance between source points, $r_{\text{nms}}$ is the non-maximum suppression radius, and $\vee$ denotes the logical OR operation. A correspondence is considered a local maximum when, for all other correspondences, either its score is higher or it is spatially distant.

The final score for each correspondence is:

$$S_i^{\text{final}} = S_i \cdot \text{IsLocalMax}_i, \tag{6}$$

**Low-scale Sampling.** We select the top-$K$ correspondences with the highest final scores, where $K$ is determined by:

$$K = \lfloor \beta \cdot \mathcal{N} \rfloor, \tag{7}$$

with $\beta$ being the sampling ratio parameter. This sampling strategy ensures that we can select a diverse set of geometrically consistent correspondences, significantly reducing the computational cost of subsequent operations while maintaining a high probability of including correct correspondences.

### 3.4 The $k$-Truss Decomposition

Following the Consensus Voting-based Low-scale Sampling Strategy presented in Sec. 3.3, we construct a new compatibility graph among the selected high-quality correspondences. The adjacency matrix $\mathbf{A}$ of this refined graph represents the pairwise geometric consistency relationships established in Sec. 3.2, but now focused on the reduced set of promising correspondences. To further identify structurally consistent subsets, we apply $k$-truss decomposition to this adjacency matrix.

**Definition 1.** *($k$-Truss Decomposition Theory). Given an undirected graph $G = (V, E)$ and an integer $k \geq 3$, the $k$-truss of $G$, denoted by $T_k(G)$, is defined as the maximal subgraph $H = (V_H, E_H)$ of $G$ where every edge $e \in E_H$ is contained in at least $(k - 2)$ triangles within $H$.*

*This formal definition captures the essential property that each edge in a $k$-truss must have strong structural support through triangle formations. In the context of correspondence graphs, a triangle represents three correspondences that are mutually consistent, which is a stronger constraint than pairwise consistency.*

**Definition 2.** *(Triangle Support). For an edge $e = (u, v) \in E$ in a graph $G = (V, E)$, the triangle support of $e$, denoted by $\sup(e, G)$, is defined as the number of triangles in $G$ that contain $e$:*

$$\sup(e, G) = |\{w \in V \setminus \{u, v\} : (u, w) \in E \wedge (v, w) \in E\}|, \tag{8}$$

*The triangle support can be efficiently computed using matrix operations. If $\mathbf{A}$ is the adjacency matrix of $G$, then:*

$$\sup((u, v), G) = (\mathbf{A}^2)_{u,v} \cdot \mathbf{A}_{u,v}, \tag{9}$$

*where $(\mathbf{A}^2)_{u,v}$ counts the number of length-2 paths between $u$ and $v$.*

**Theorem 1.** *Let $(p_i, p_j, p_k)$ denote a triplet of point correspondences, and let $\Delta D_{ijk}$ represent the deviation vector of triangle edge lengths between the source and target point clouds. Under a rigid transformation with Gaussian noise, for any threshold $\epsilon > 0$,*

$$\mathbb{P}\left(\|\Delta D_{ijk}\|_F < \epsilon \mid correct\right) \gg \mathbb{P}\left(\|\Delta D_{ijk}\|_F < \epsilon \mid incorrect\right),$$

*where $\mathbb{P}(\cdot)$ denotes the probability, $\|\Delta D_{ijk}\|_F$ denotes the Frobenius norm of the deviation vector, and $\gg$ indicates "significantly greater than." This inequality states that the probability of triangle relationships being preserved under rigid transformation is significantly higher for correct correspondences than for incorrect ones.*

*Therefore, the $k$-truss, which requires each edge to be supported by at least $k - 2$ triangles, significantly enhances robustness in correspondence selection by leveraging higher-order structural consistency.*

*Please see Appendix A.2 for detailed proof and derivations.*

**Matrix-Based Implementation.** The $k$-truss decomposition operates on the adjacency matrix derived from the sampled correspondences. The algorithm proceeds by computing the triangle support for each edge:

$$\mathbf{T} = \mathbf{A}^2 \odot \mathbf{A}, \tag{10}$$

where $\odot$ denotes the Hadamard (element-wise) product.

We then identify valid edges that satisfy the $k$-truss criterion [9]:

$$\mathbf{E}_{\text{valid}} = (\mathbf{T} \geq (k - 2)), \tag{11}$$

For each vertex $i$, we extract its neighborhood connected by valid edges:

$$\mathcal{N}_i = \{j \in V : \mathbf{E}_{\text{valid}}(i, j) = 1\}, \tag{12}$$

Vertices with neighborhoods of sufficient size ($|\mathcal{N}_i| \geq k$) form clusters in the $k$-truss decomposition.

**Computational Complexity Analysis of $k$-Truss Decomposition.** The $k$-truss decomposition has a time complexity of $O(m^{1.5})$, which is much more efficient than exponential-time clique-based methods and remains practical for large-scale graphs. For more details, please refer to Appendix A.3.

### 3.5 Hypothesis Generation and Evaluation

Each $k$-truss subgraph filtered from the previous step represents a structurally robust set of correspondences. By applying the SVD algorithm to each $k$-truss subgraph, we can obtain a set of 6-DoF pose hypotheses.

**Centrality-weighted SVD.** Transformation estimation of correspondences is implemented using weighted SVD [48, 29, 5, 7]. We assign weights to correspondences based on their centrality values within the $k$-truss subgraph. Our weighting scheme follows established graph-based PCR methods by deriving weights from spectral analysis [24]. We compute the eigendecomposition of the $k$-truss subgraph's compatibility matrix and use the principal eigenvector elements as correspondence weights in our weighted SVD. This method leverages the structural importance of each correspondence to improve transformation accuracy.

The final goal of our method is to estimate the optimal 6-DoF rigid transformation (composed of a rotation pose $\mathbf{R}^* \in SO(3)$ and a translation pose $\mathbf{t}^* \in \mathbb{R}^3$) that maximizes our spatial distribution score function:

$$(\mathbf{R}^*, \mathbf{t}^*) = \arg\max_{\mathbf{R}, \mathbf{t}} \mathrm{SDS}(\mathbf{P}, \mathbf{Q}, \mathbf{R}, \mathbf{t}), \tag{13}$$

where SDS represents our spatial distribution score function defined as:

$$\mathrm{SDS}(\mathbf{P}, \mathbf{Q}, \mathbf{R}, \mathbf{t}) = \rho_{\mathrm{inlier}} \cdot \sqrt{\rho_{\mathrm{coverage}}} \cdot \rho_{\mathrm{error}}, \tag{14}$$

with the individual components:

$$\rho_{\mathrm{inlier}} = \frac{|\mathcal{I}|}{|\mathbf{P}|}, \tag{15}$$

$$\rho_{\mathrm{coverage}} = \frac{\prod_{d \in \{x,y,z\}} \mathrm{range}_d(\mathcal{I})}{\prod_{d \in \{x,y,z\}} \mathrm{range}_d(\mathbf{P})}, \tag{16}$$

$$\rho_{\mathrm{error}} = 1 - \frac{\frac{1}{|\mathcal{I}|} \sum_{i \in \mathcal{I}} \|\mathbf{R}\mathbf{p}_i + \mathbf{t} - \mathbf{q}_i\|}{\tau}, \tag{17}$$

where $\mathcal{I} = \{i : \|\mathbf{R}\mathbf{p}_i + \mathbf{t} - \mathbf{q}_i\| < \tau\}$ is the set of inlier indices, $\tau$ is the inlier threshold, $\mathrm{range}_d$ computes the coordinate range along dimension $d$, $\mathbf{p}_i \in \mathbf{P}$ and $\mathbf{q}_i \in \mathbf{Q}$ are corresponding points from source and target point clouds.

Unlike conventional metrics such as MAE [48] or inlier count [5], our SDS function comprehensively evaluates both alignment accuracy and spatial distribution quality of inliers. The $\rho_{\mathrm{inlier}}$ term measures the proportion of correctly aligned points, $\rho_{\mathrm{coverage}}$ evaluates how well the inliers span the original point cloud volume, and $\rho_{\mathrm{error}}$ assesses the precision of alignment among inlier points. This balanced evaluation method effectively prevents selecting transformations with clustered inliers in limited regions and promotes transformations with well-distributed inliers across the entire object. The best hypothesis according to this comprehensive scoring function is selected to perform the final 3D registration.

## 4 Experiment

### 4.1 Experimental Setup

**Datasets.** For outdoor scenarios, we evaluate our method on the KITTI dataset [15]. Following the protocol established in [1, 5, 48], we select 555 point cloud pairs from sequences 8 to 10 for testing. For indoor environments, we conduct experiments on the 3DMatch dataset [45] and the more challenging 3DLoMatch dataset [17], where point cloud pairs have less than $30\%$ overlap. To further assess the robustness and generalization capability of our approach, we also perform experiments on the Bunny model from the Stanford 3D Scanning Repository [10].

**Evaluation Criteria.** We use rotation error (RE), translation error (TE), and registration recall (RR) as the main evaluation metrics. Following [4, 39, 48], registration is considered successful if the results on the 3DMatch and 3DLoMatch datasets satisfy $\text{RE} \leq 15°$ and $\text{TE} \leq 30\text{cm}$, or on the KITTI dataset $\text{RE} \leq 5°$ and $\text{TE} \leq 60\text{cm}$. The mean rotation error and mean translation error are computed only on successfully registered pairs. The registration accuracy of a dataset is defined as the ratio of successfully registered pairs to the total number of pairs.

**Implementation details.** Our method is implemented in PyTorch. For the 3DMatch, 3DLoMatch, and KITTI datasets, we use Fast Point Feature Histograms (FPFH) [31] and Fully Convolutional Geometric Features (FCGF) [8] as descriptors to generate initial correspondences. Following [4, 39, 21], the Bunny model is downsampled to $N_c$ points and resized to fit a $[0, 1]^3$ cube, creating the source point cloud $\mathcal{P}$. To generate the target point cloud $\mathcal{Q}$, a random transformation $(\mathbf{R}, \mathbf{t})$ is applied to $\mathcal{P}$ and then Gaussian noise $\boldsymbol{\epsilon}_i \sim \mathcal{N}(0, \sigma^2 \mathbf{I}_3)$ is added. A pair of the original and moved points defines an inlier. The inliers are contaminated with outliers generated by random transformations. Detailed computational complexity analysis is provided in the Appendix A.3. All experiments are conducted on an AMD Ryzen 9 5950X CPU and a single NVIDIA RTX 3090 GPU.

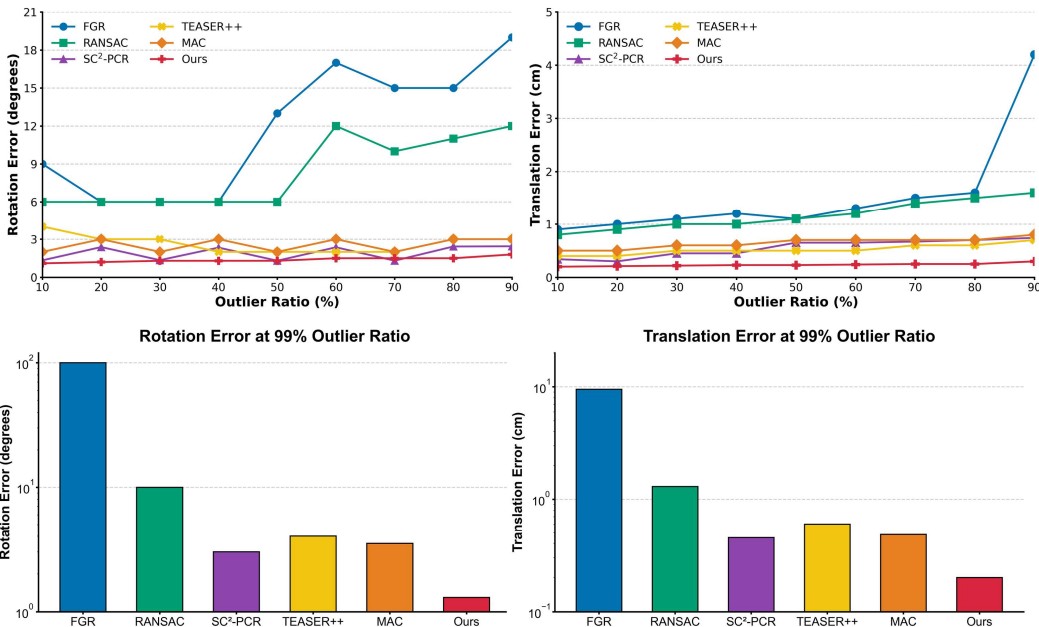

Figure 3: **Outlier robustness evaluation on the synthetic dataset.** The first row shows the rotation and translation errors of each method as the outlier ratio on the Bunny model increases from 10% to 90%. The second row compares the rotation and translation errors of different methods at an outlier ratio of 99%.

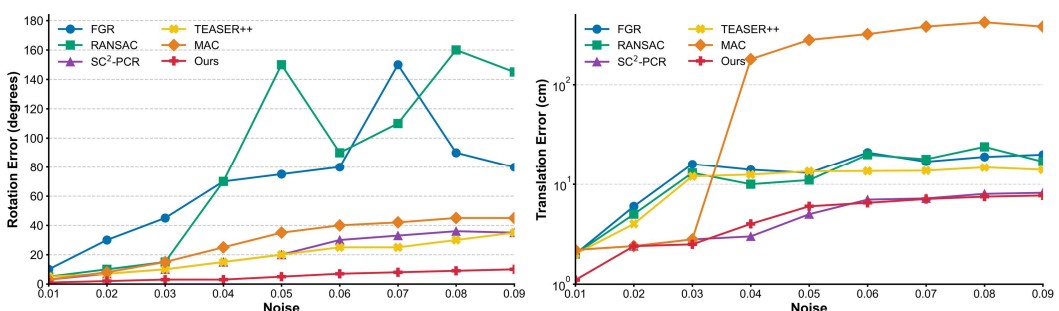

Figure 4: **Noise robustness evaluation on the synthetic dataset.** Comparison of rotation and translation errors on the Bunny model as the noise standard deviation increases from 0.01 to 0.09.

## 4.2 Robustness to Outliers and Noise on Synthetic Data

On the synthetic dataset, we conduct experiments using the Bunny model. The outlier ratio is varied from 10% to 90% to systematically evaluate the robustness of each method under high outlier rates. The Bunny model is downsampled to $N_c = 500$ and Gaussian noise with zero mean and standard deviation $\sigma = 0.01$ is added. For each outlier ratio, 100 independent trials are performed, and the mean rotation error (RE) and translation error (TE) are recorded. We also test an extreme case with an outlier ratio of up to 99% (please see Fig. 3, second row). Our method is compared against state-of-the-art traditional approaches [51, 14, 5, 39, 48]. Results show that traditional methods such as FGR and RANSAC exhibit rapidly increasing errors as the outlier ratio rises, while our method consistently achieves the best robustness and registration accuracy across all outlier levels.

Table 1: Results on KITTI dataset [15] using FPFH [31] and FCGF [8] descriptors.

| | FPFH | | | FCGF | | | |
| --- | --- | --- | --- | --- | --- | --- | --- |
| | RR(%)↑ | RE(°)↓ | TE(cm)↓ | RR(%)↑ | RE(°)↓ | TE(cm)↓ | Time(s) |
| *i) Traditional* | | | | | | | |
| FGR [51] | 5.23 | 0.86 | 43.84 | 89.54 | 0.46 | 25.72 | 3.88 |
| RANSAC [14] | 74.41 | 1.55 | 30.20 | 80.36 | 0.73 | 26.79 | 5.43 |
| TEASER++ [39] | 91.17 | 1.03 | 17.98 | 95.51 | 0.33 | 22.38 | 0.03 |
| SC$^2$-PCR [5] | 99.46 | 0.35 | 7.87 | 98.02 | 0.33 | 20.69 | 0.31 |
| MAC [48] | 97.66 | 0.41 | 8.61 | 97.84 | 0.34 | 19.34 | 3.29 |
| TR-DE [4] | 96.76 | 0.90 | 15.63 | 98.20 | 0.38 | 18.00 | - |
| TEAR [18] | 99.10 | 0.39 | 8.62 | - | - | - | - |
| Jiang et al. [21] | 99.56 | **0.34** | 7.85 | 98.20 | **0.32** | 20.73 | 0.54 |
| *ii) Deep learned* | | | | | | | |
| DGR [7] | 77.12 | 1.64 | 33.10 | 96.90 | 0.34 | 21.70 | 2.29 |
| PointDSC [1] | 98.92 | 0.38 | 8.35 | 97.84 | 0.33 | 20.32 | 0.45 |
| VBReg [20] | 98.92 | 0.45 | 8.41 | 98.02 | **0.32** | 20.91 | 0.24 |
| Ours | **99.64** | 0.43 | **5.31** | **99.10** | 0.59 | **11.06** | 0.21 |

We further evaluate the robustness of each method under different noise levels, as shown in Fig. 4. Specifically, we increase the standard deviation of Gaussian noise from $\sigma = 0.01$ to $\sigma = 0.1$ to systematically assess algorithm performance. Experimental results indicate that, as the noise level increases, the translation error of the clique-based MAC [48] method rises significantly, while our triangle-based method is barely affected. The bundled structure of triangles effectively captures the key skeleton of the point cloud and resists noise interference. As a result, our method consistently achieves the lowest rotation and translation errors under high noise conditions, demonstrating superior robustness.

Table 2: Comparison results on 3DMatch [45] using FPFH [31] and FCGF [8] descriptors.

| | FPFH | | | FCGF | | | |
| --- | --- | --- | --- | --- | --- | --- | --- |
| | RR(%)↑ | RE(°)↓ | TE(cm)↓ | RR(%)↑ | RE(°)↓ | TE(cm)↓ | Time(s) |
| *i) Traditional* | | | | | | | |
| FGR [51] | 40.91 | 4.96 | 10.25 | 78.93 | 2.90 | 8.41 | 0.89 |
| RANSAC [14] | 66.10 | 3.95 | 11.03 | 91.44 | 2.69 | 8.38 | 2.86 |
| TEASER++ [39] | 75.48 | 2.48 | 7.31 | 85.71 | 2.73 | 8.66 | 0.03 |
| SC$^2$-PCR [5] | 83.90 | 2.12 | 6.69 | 93.16 | 2.06 | 6.53 | 0.12 |
| MAC [48] | 83.90 | 2.11 | 6.80 | 93.72 | 2.07 | 6.52 | 5.54 |
| FastMAC [49] | 82.87 | 2.15 | 6.73 | 92.67 | 2.00 | 6.47 | 0.11 |
| Jiang et al. [21] | 83.92 | 2.12 | 6.64 | 93.28 | 2.04 | 6.48 | 0.36 |
| *ii) Deep learned* | | | | | | | |
| DGR [7] | 32.84 | 2.45 | 7.53 | 88.85 | 2.28 | 7.02 | 1.53 |
| PointDSC [1] | 72.95 | 2.18 | 6.45 | 91.87 | 2.10 | 6.54 | 0.10 |
| VBReg [20] | 82.57 | 2.14 | 6.77 | 93.53 | 2.04 | 6.49 | 0.20 |
| Ours | **84.70** | **1.80** | **6.22** | **93.84** | **1.70** | **6.13** | 0.20 |

## 4.3 Experimental Results on the KITTI Dataset

We conduct experiments on the KITTI dataset [15] to evaluate the potential of our algorithm in real outdoor scenarios. Table 1 presents the results using FPFH [31] and FCGF [8] descriptors for initial correspondence generation. We compare our method with leading traditional [51, 14, 39, 5, 48, 4, 18, 21] and learning-based approaches [7, 1, 20]. Following [5, 48], the mean rotation error (RE) and mean translation error (TE) are calculated only on successfully registered pairs. As shown in Table 1, our method achieves the highest recall (RR) and lowest TE with both FPFH and FCGF descriptors. Moreover, our method demonstrates superior efficiency at comparable registration accuracy. These results confirm the robustness of our method for registering sparse and non-uniform outdoor point clouds. Additional visualizations are provided in the Appendix A.10.

Table 3: Comparison results on 3DLoMatch [17] using FPFH [31] and FCGF [8] descriptors.

| | FPFH | | | FCGF | | |
|---|---|---|---|---|---|---|
| | RR(%)↑ | RE(°)↓ | TE(cm)↓ | RR(%)↑ | RE(°)↓ | TE(cm)↓ |
| *i) Traditional* | | | | | | |
| RANSAC [14] | 19.83 | 4.67 | 10.32 | 37.60 | 4.28 | 11.04 |
| TEASER++ [39] | 35.15 | 4.38 | 10.96 | 46.76 | 4.12 | 12.89 |
| SC$^2$-PCR [5] | 35.93 | 4.26 | 10.86 | 58.73 | 3.80 | 10.44 |
| MAC [48] | 40.88 | 3.66 | 9.45 | 59.85 | 3.50 | 9.75 |
| FastMAC [49] | 38.46 | 4.04 | 10.47 | 58.23 | 3.80 | 10.81 |
| *ii) Deep learned* | | | | | | |
| PointDSC [1] | 27.91 | 4.27 | 10.45 | 56.20 | 3.87 | 10.48 |
| VBReg [20] | 30.83 | 4.38 | 10.92 | 58.30 | 3.58 | **9.72** |
| Ours | **43.96** | **2.89** | **8.93** | **61.64** | **3.30** | **9.72** |

## 4.4 Experimental Results on the 3DMatch and 3DLoMatch Datasets

We conducted systematic comparative experiments on the 3DMatch dataset with overlap ratios exceeding 30%. The left and right columns of Table 2 show the registration performance using FPFH and FCGF descriptors, respectively. With the handcrafted FPFH descriptor, our method achieves the highest recall (RR), outperforming both traditional and learning-based approaches. Using the FCGF descriptor, our method surpasses all state-of-the-art baselines on every evaluation metric. Compared to the MAC method, our method improves RR by 0.56%. More importantly, it reduces the average rotation error (RE) and average translation error (TE) by about 16.7% and 5.4%, respectively. This demonstrates superior overall performance. Qualitative results are shown in Fig. 5 and Appendix A.10. Our method remains robust even in challenging scenarios with ambiguous features or unclear local structures. It achieves alignment results that are close to the ground truth. These findings strongly validate the robustness and generalization ability of our method on diverse and complex point cloud data.

As shown in Table 3, we systematically evaluated our algorithm on the 3DLoMatch dataset for low-overlap registration. We compared our method with several leading traditional and deep learning approaches, using both FPFH and FCGF descriptors. Our method consistently delivers superior recall rates and reduced error metrics, validating its exceptional robustness and versatility even in challenging low-overlap scenarios. Qualitative results in Fig. 5 and Appendix A.10 further illustrate that our method remains effective even when local structures are ambiguous.

## 4.5 PointTruss Integration with Deep Learning Methods on 3DLoMatch

We have conducted experiments combining PointTruss with recent deep learning methods [29, 41] on the challenging 3DLoMatch dataset.

PointTruss successfully enhances both GeoTransformer (+4.5% recall) and PareNet (+1.70% recall) on the challenging 3DLoMatch dataset. The consistent improvements across different learned features validate PointTruss's compatibility with modern deep learning pipelines. Moreover, when integrated with GeoTransformer, PointTruss achieves performance comparable to MAC while being more computationally efficient. These results demonstrate that PointTruss not only works as a standalone

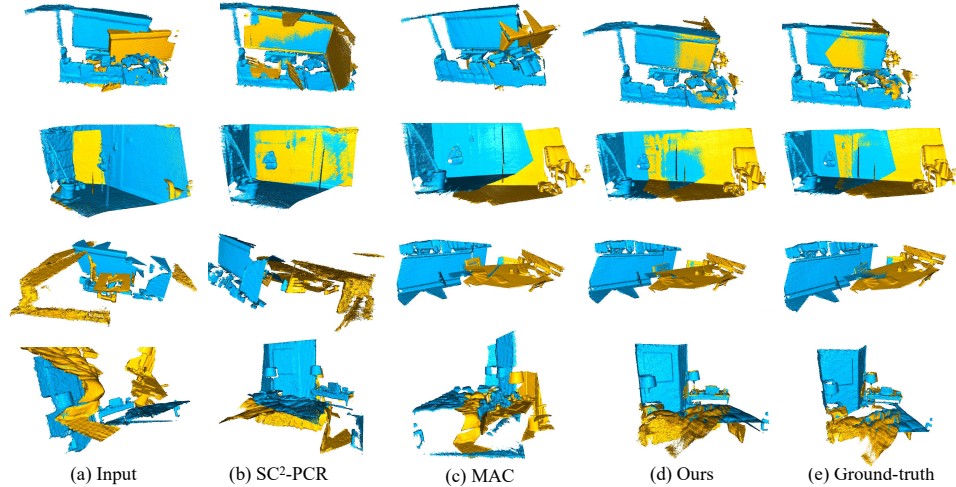

| (a) Input | (b) SC²-PCR | (c) MAC | (d) Ours | (e) Ground-truth |

Figure 5: **Qualitative comparisons on the 3DMatch and 3DLoMatch datasets.** The first and second rows correspond to 3DMatch, and the third and fourth rows correspond to 3DLoMatch.

Table 4: PointTruss Integration with Deep Learning Methods on 3DLoMatch

| Method | Registration Recall |
|---|---|
| GeoTransformer | 75.0% |
| GeoTransformer + MAC | 78.9% |
| **GeoTransformer + PointTruss** | **79.5%** |
| PareNet | 80.5% |
| PareNet + MAC | 81.5% |
| **PareNet + PointTruss** | **82.2%** |

method but also serves as an effective drop-in replacement for traditional robust estimators in deep learning pipelines, providing consistent improvements across different feature extractors.

## 5   Conclusion

In this work, we introduce the $k$-truss from graph theory to the point cloud registration and use triangle support as a key constraint. We first perform consensus voting-based low-scale sampling on the input correspondences to construct a compatibility graph. Based on this, we propose a heuristic method that applies $k$-truss decomposition with triangle support constraints to obtain several $k$-truss subgraphs. Each candidate subgraph is then processed by weighted SVD, and we use a designed spatial distribution score to evaluate the spatial coverage and uniformity of inliers, selecting the best transformation hypothesis. Our method is efficient and simple, leveraging triangles as minimal rigid planar structures and exploiting their strong structural binding. Experimental results on indoor, outdoor, and object-level point clouds show that our algorithm achieves state-of-the-art registration accuracy while maintaining high efficiency. The method is robust to large numbers of outliers and low-overlap scenarios. Limitations and broader impacts are discussed in Appendix A.7.

## 6   Acknowledgements

This work is supported by the National Natural Science Foundation of China (62036006, 62276200) and the Innovation Capability Support Plan of Shaanxi Province (2023KJXX-144).

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

# A  Appendix

In the appendix, we first provide rigorous definitions of evaluation metrics (Sec. A.1). We then present the Triangle Relation Stability Theorem for point cloud registration (Sec. A.2). A computational complexity analysis and comparison of dense subgraph algorithms is given in Sec. A.3. We further describe pseudocode for key algorithmic components (Sec. A.4) and provide specific hyper-parameter selections for reference (Sec. A.5). Additionally, we conduct ablation studies of each component (Sec. A.6) and discuss the limitations and scalability of our method (Sec. A.7 and Sec. A.8). We offer detailed information on public datasets (Sec. A.9) along with their visualization results (Sec. A.10). We also present an ablation study on graph sampling and the k-value (Sec. A.11).

## A.1  The rigorous definitions of Evaluation Metrics

**Rotation Error (RE)**    For a given point cloud pair, the rotation error measures the angular difference between the estimated rotation $\mathbf{R}$ and the ground truth rotation $\mathbf{R}_{gt}$, computed as:

$$\text{RE} = \arccos \frac{\text{trace}(\mathbf{R}_{gt}\mathbf{R}^T) - 1}{2} \tag{18}$$

**Translation Error (TE)**    The translation error measures the Euclidean distance between the estimated translation $\mathbf{t}$ and the ground truth translation $\mathbf{t}_{gt}$:

$$\text{TE} = \|\mathbf{t} - \mathbf{t}_{gt}\|_2 \tag{19}$$

**Registration Recall (RR)**    Registration recall measures the percentage of successfully registered point cloud pairs over all pairs in the dataset. A registration is considered successful if:

$$\text{RR}_{\text{3DMatch\&3DLoMatch}} = \frac{1}{N} \sum_{i=1}^{N} [\text{RE}_i < 15° \wedge \text{TE}_i < 30\,\text{cm}] \tag{20}$$

$$\text{RR}_{\text{KITTI}} = \frac{1}{N} \sum_{i=1}^{N} [\text{RE}_i < 5° \wedge \text{TE}_i < 60\,\text{cm}] \tag{21}$$

**Mean Rotation and Translation Errors**    The mean rotation and translation errors are computed only over the successfully registered point cloud pairs:

$$\text{Mean RE} = \frac{1}{|N'|} \sum_{i \in N'} \text{RE}_i \tag{22}$$

$$\text{Mean TE} = \frac{1}{|N'|} \sum_{i \in N'} \text{TE}_i \tag{23}$$

where $N' = \{i \mid \text{RE}_i < \tau_{\text{RE}} \wedge \text{TE}_i < \tau_{\text{TE}}\}$ denotes the set of successfully registered pairs, with $\tau_{\text{RE}} = 15°$ and $\tau_{\text{TE}} = 30\,\text{cm}$ for 3DMatch and 3DLoMatch, and $\tau_{\text{RE}} = 5°$ and $\tau_{\text{TE}} = 60\,\text{cm}$ for KITTI.

## A.2  Triangle Relation Stability Theorem Based Point Cloud Registration

The PointTruss registration framework introduces a novel perspective by leveraging the $k$-truss from graph theory to establish robust correspondence patterns in point cloud registration. At the core of $k$-truss lies the concept of triangle support—each edge in a $k$-truss is contained in at least k-2 triangles, providing exceptional structural stability against perturbations. This property is particularly advantageous in point cloud registration, where noise, outliers, and partial visibility are common challenges.

Before developing the algorithmic components of PointTruss, it is essential to establish a rigorous theoretical foundation that quantifies how triangle relations behave under noise and rigid transformations. Specifically, we need to mathematically prove why triangle relations remain stable for

correct correspondences while exhibiting significant deviations for incorrect ones. This theoretical foundation addresses several critical questions:

1. How do triangle relations (edge lengths) change under noise perturbation?

2. What statistical properties characterize these changes?

3. Under what conditions can we reliably distinguish between correct and incorrect point correspondences based on triangle relation stability?

4. Why does the triangle-supported $k$-truss provide a reliable foundation for robust registration?

The following theorem establishes the statistical properties of triangle relations under noise, providing the theoretical underpinning for the PointTruss registration framework. By demonstrating that correct correspondences maintain stable triangle relations with high probability, this analysis justifies using triangle-based constraints as a core mechanism for robust point cloud registration.

Let $\mathcal{P} = \{p_i\}_{i=1}^N$ denote the source point cloud and $\mathcal{Q} = \{q_i\}_{i=1}^N$ denote the target point cloud after rigid transformation and noise perturbation, where $p_i, q_i \in \mathbb{R}^3$. The rigid transformation is defined as $T = (R, t)$, where $R \in SO(3)$ is a rotation matrix satisfying $R^\top R = I$ and $\det(R) = 1$, and $t \in \mathbb{R}^3$ is a translation vector.

The noise-contaminated point cloud model can be expressed as:

$$q_i = Rp_i + t + \eta_i, \quad \eta_i \sim \mathcal{N}(0, \sigma^2 I) \tag{24}$$

where $\eta_i$ represents independent and identically distributed Gaussian noise with zero mean and covariance matrix $\sigma^2 I$.

For any pair of points $p_i$ and $p_j$ in the source point cloud, their Euclidean distance is preserved under rigid transformation. Let $d_{ij} = \|p_i - p_j\|$ denote the distance between points $p_i$ and $p_j$. In the absence of noise, the distance between the corresponding transformed points $q_i$ and $q_j$ is:

$$\begin{aligned}
\|q_i - q_j\| &= \|Rp_i + t - (Rp_j + t)\| \\
&= \|R(p_i - p_j)\| \\
&= \sqrt{(R(p_i - p_j))^\top (R(p_i - p_j))} \\
&= \sqrt{(p_i - p_j)^\top R^\top R(p_i - p_j)} \\
&= \sqrt{(p_i - p_j)^\top I(p_i - p_j)} \\
&= \|p_i - p_j\| = d_{ij}
\end{aligned} \tag{25}$$

This distance preservation property is a fundamental characteristic of rigid transformations.

For any triplet of points $(p_i, p_j, p_k)$ in the source point cloud, we define the triangle relation matrix $D_{ijk}$ as:

$$D_{ijk} = [\|p_i - p_j\| \quad \|p_i - p_k\| \quad \|p_j - p_k\|] \tag{26}$$

Similarly, for the corresponding points in the target point cloud, the triangle relation matrix is:

$$D'_{ijk} = [\|q_i - q_j\| \quad \|q_i - q_k\| \quad \|q_j - q_k\|] \tag{27}$$

Under noise-free rigid transformation, $D_{ijk} = D'_{ijk}$, reflecting the invariance of triangle relations under rigid transformations.

In the presence of noise, the distance between two points in the target point cloud becomes:

$$\begin{aligned}
\|q_i - q_j\| &= \|Rp_i + t + \eta_i - (Rp_j + t + \eta_j)\| \\
&= \|R(p_i - p_j) + (\eta_i - \eta_j)\| \\
&= \|p_i - p_j + R^\top(\eta_i - \eta_j)\| \\
&= \|p_i - p_j + Z_{ij}\|
\end{aligned} \tag{28}$$

where $Z_{ij} = R^\top(\eta_i - \eta_j) \sim \mathcal{N}(0, 2\sigma^2 I)$ represents the transformed noise difference.

The squared distance between noisy points can be expanded as:

$$\begin{aligned}
\|q_i - q_j\|^2 &= \|p_i - p_j + Z_{ij}\|^2 \\
&= (p_i - p_j + Z_{ij})^\top(p_i - p_j + Z_{ij}) \\
&= (p_i - p_j)^\top(p_i - p_j) + 2(p_i - p_j)^\top Z_{ij} + Z_{ij}^\top Z_{ij} \\
&= \|p_i - p_j\|^2 + 2(p_i - p_j)^\top Z_{ij} + \|Z_{ij}\|^2
\end{aligned} \tag{29}$$

Taking the expectation:

$$\begin{aligned}
\mathbb{E}[\|q_i - q_j\|^2] &= \mathbb{E}[\|p_i - p_j\|^2 + 2(p_i - p_j)^\top Z_{ij} + \|Z_{ij}\|^2] \\
&= \|p_i - p_j\|^2 + \mathbb{E}[2(p_i - p_j)^\top Z_{ij}] + \mathbb{E}[\|Z_{ij}\|^2]
\end{aligned} \tag{30}$$

Since $\mathbb{E}[Z_{ij}] = 0$, the middle term vanishes:

$$\mathbb{E}[2(p_i - p_j)^\top Z_{ij}] = 2(p_i - p_j)^\top \mathbb{E}[Z_{ij}] = 0 \tag{31}$$

For the third term, $\|Z_{ij}\|^2$ follows a chi-squared distribution with 3 degrees of freedom and scaling factor $2\sigma^2$. The expected value of a chi-squared random variable is its degrees of freedom multiplied by the scaling factor:

$$\mathbb{E}[\|Z_{ij}\|^2] = 3 \cdot 2\sigma^2 = 6\sigma^2 \tag{32}$$

Therefore:

$$\mathbb{E}[\|q_i - q_j\|^2] = d_{ij}^2 + 6\sigma^2 \tag{33}$$

To calculate the expected value of $\|q_i - q_j\|$, we apply a first-order Taylor expansion of $f(x) = \sqrt{x}$ around $x = d_{ij}^2$:

$$\begin{aligned}
f(x) &\approx f(d_{ij}^2) + f'(d_{ij}^2)(x - d_{ij}^2) \\
&= \sqrt{d_{ij}^2} + \frac{1}{2\sqrt{d_{ij}^2}}(x - d_{ij}^2) \\
&= d_{ij} + \frac{1}{2d_{ij}}(x - d_{ij}^2)
\end{aligned} \tag{34}$$

For this approximation to be valid, we require $\sigma \ll d_{ij}$, ensuring that higher-order terms (of order $\frac{\sigma^4}{d_{ij}^3}$ and beyond) remain negligible.

Substituting $x = \|q_i - q_j\|^2$ with $\mathbb{E}[x] = d_{ij}^2 + 6\sigma^2$:

$$\mathbb{E}[\|q_i - q_j\|] \approx d_{ij} + \frac{1}{2d_{ij}}(\mathbb{E}[\|q_i - q_j\|^2] - d_{ij}^2)$$

$$= d_{ij} + \frac{1}{2d_{ij}}(d_{ij}^2 + 6\sigma^2 - d_{ij}^2)$$

$$= d_{ij} + \frac{1}{2d_{ij}} \cdot 6\sigma^2 \tag{35}$$

$$= d_{ij} + \frac{3\sigma^2}{d_{ij}}$$

Defining the distance deviation as $\Delta_{ij} = \|q_i - q_j\| - d_{ij}$, its expected value is:

$$\mathbb{E}[\Delta_{ij}] = \mathbb{E}[\|q_i - q_j\|] - d_{ij} \approx \frac{3\sigma^2}{d_{ij}} \tag{36}$$

For more precise analysis, we decompose the noise vector $Z_{ij}$ into components parallel and perpendicular to the direction vector $u_{ij} = \frac{p_i - p_j}{d_{ij}}$:

$$Z_{ij} = Z_{ij,u} u_{ij} + Z_{ij,\perp} \tag{37}$$

where $Z_{ij,u} = Z_{ij} \cdot u_{ij}$ is the projection of noise along $u_{ij}$, and $Z_{ij,\perp}$ is the perpendicular component. Using this decomposition, the distance can be more precisely approximated as:

$$\|q_i - q_j\| = \|d_{ij} u_{ij} + Z_{ij}\|$$

$$= \|d_{ij} u_{ij} + Z_{ij,u} u_{ij} + Z_{ij,\perp}\|$$

$$= \|(d_{ij} + Z_{ij,u}) u_{ij} + Z_{ij,\perp}\|$$

$$\approx (d_{ij} + Z_{ij,u})\sqrt{1 + \frac{\|Z_{ij,\perp}\|^2}{(d_{ij} + Z_{ij,u})^2}} \tag{38}$$

$$\approx (d_{ij} + Z_{ij,u})\left(1 + \frac{1}{2}\frac{\|Z_{ij,\perp}\|^2}{(d_{ij} + Z_{ij,u})^2}\right)$$

For $\|Z_{ij,u}\| \ll d_{ij}$, we can further approximate:

$$\|q_i - q_j\| \approx (d_{ij} + Z_{ij,u})\left(1 + \frac{1}{2}\frac{\|Z_{ij,\perp}\|^2}{d_{ij}^2}\right)$$

$$\approx d_{ij} + Z_{ij,u} + \frac{\|Z_{ij,\perp}\|^2}{2d_{ij}} \tag{39}$$

Since $\|Z_{ij,\perp}\|^2 = \|Z_{ij}\|^2 - Z_{ij,u}^2$, we have:

$$\|q_i - q_j\| \approx d_{ij} + Z_{ij,u} + \frac{\|Z_{ij}\|^2 - Z_{ij,u}^2}{2d_{ij}} \tag{40}$$

Therefore, the deviation can be expressed as:

$$\Delta_{ij} \approx Z_{ij,u} + \frac{\|Z_{ij}\|^2 - Z_{ij,u}^2}{2d_{ij}} \tag{41}$$

The variance of $\Delta_{ij}$ can be decomposed as:

$$\text{Var}(\Delta_{ij}) = \text{Var}\left(Z_{ij,u} + \frac{\|Z_{ij}\|^2 - Z_{ij,u}^2}{2d_{ij}}\right)$$
$$= \text{Var}(Z_{ij,u}) + \text{Var}\left(\frac{\|Z_{ij}\|^2 - Z_{ij,u}^2}{2d_{ij}}\right) + 2\text{Cov}\left(Z_{ij,u}, \frac{\|Z_{ij}\|^2 - Z_{ij,u}^2}{2d_{ij}}\right) \tag{42}$$

Since $Z_{ij} \sim \mathcal{N}(0, 2\sigma^2 I)$, we have $Z_{ij,u} \sim \mathcal{N}(0, 2\sigma^2)$, thus:

$$\text{Var}(Z_{ij,u}) = 2\sigma^2 \tag{43}$$

For the second term, $\|Z_{ij}\|^2 \sim 2\sigma^2\chi_3^2$ with variance $12\sigma^4$, and $Z_{ij,u}^2 \sim 2\sigma^2\chi_1^2$ with variance $8\sigma^4$. The covariance between them is $4\sigma^4$. Therefore:

$$\begin{aligned}
\text{Var}\left(\frac{\|Z_{ij}\|^2 - Z_{ij,u}^2}{2d_{ij}}\right) &= \frac{1}{4d_{ij}^2}\text{Var}(\|Z_{ij}\|^2 - Z_{ij,u}^2) \\
&= \frac{1}{4d_{ij}^2}(\text{Var}(\|Z_{ij}\|^2) + \text{Var}(Z_{ij,u}^2) - 2\text{Cov}(\|Z_{ij}\|^2, Z_{ij,u}^2)) \\
&= \frac{1}{4d_{ij}^2}(12\sigma^4 + 8\sigma^4 - 2 \cdot 4\sigma^4) \\
&= \frac{1}{4d_{ij}^2}(20\sigma^4 - 8\sigma^4) \\
&= \frac{1}{4d_{ij}^2} \cdot 12\sigma^4 \\
&= \frac{3\sigma^4}{d_{ij}^2}
\end{aligned} \tag{44}$$

The third term vanishes due to the independence between $Z_{ij,u}$ and $\|Z_{ij,\perp}\|^2$:

$$\text{Cov}\left(Z_{ij,u}, \frac{\|Z_{ij}\|^2 - Z_{ij,u}^2}{2d_{ij}}\right) = \frac{1}{2d_{ij}}\text{Cov}(Z_{ij,u}, \|Z_{ij,\perp}\|^2) = 0 \tag{45}$$

Hence, the variance of distance deviation is:

$$\text{Var}(\Delta_{ij}) \approx 2\sigma^2 + \frac{3\sigma^4}{d_{ij}^2} \tag{46}$$

For the triplet $(p_i, p_j, p_k)$, we need to calculate the covariance between deviations $\Delta_{ij}$ and $\Delta_{ik}$. The key insight is that $Z_{ij} = R^\top(\eta_i - \eta_j)$ and $Z_{ik} = R^\top(\eta_i - \eta_k)$ share the noise component $\eta_i$.

The cross-covariance matrix is:

$$\begin{aligned}
\mathbb{E}[Z_{ij}Z_{ik}^\top] &= \mathbb{E}[R^\top(\eta_i - \eta_j)(\eta_i - \eta_k)^\top R] \\
&= R^\top \mathbb{E}[(\eta_i - \eta_j)(\eta_i - \eta_k)^\top]R
\end{aligned} \tag{47}$$

Expanding the expectation:

$$\begin{aligned}
\mathbb{E}[(\eta_i - \eta_j)(\eta_i - \eta_k)^\top] &= \mathbb{E}[\eta_i\eta_i^\top - \eta_i\eta_k^\top - \eta_j\eta_i^\top + \eta_j\eta_k^\top] \\
&= \mathbb{E}[\eta_i\eta_i^\top] - \mathbb{E}[\eta_i\eta_k^\top] - \mathbb{E}[\eta_j\eta_i^\top] + \mathbb{E}[\eta_j\eta_k^\top]
\end{aligned} \tag{48}$$

Due to independence of noise vectors, only $\mathbb{E}[\eta_i\eta_i^\top] = \sigma^2 I$ is non-zero. Therefore:

$$\mathbb{E}[(\eta_i - \eta_j)(\eta_i - \eta_k)^\top] = \sigma^2 I - 0 - 0 + 0 = \sigma^2 I$$
$$\mathbb{E}[Z_{ij} Z_{ik}^\top] = R^\top \sigma^2 I R = \sigma^2 I \tag{49}$$

This leads to the covariance between projections:

$$\begin{aligned}
\mathrm{Cov}(Z_{ij,u}, Z_{ik,u}) &= \mathbb{E}[Z_{ij,u} Z_{ik,u}] - \mathbb{E}[Z_{ij,u}]\mathbb{E}[Z_{ik,u}] \\
&= \mathbb{E}[(Z_{ij} \cdot u_{ij})(Z_{ik} \cdot u_{ik})] \\
&= \mathbb{E}[u_{ij}^\top Z_{ij} Z_{ik}^\top u_{ik}] \\
&= u_{ij}^\top \mathbb{E}[Z_{ij} Z_{ik}^\top] u_{ik} \\
&= u_{ij}^\top \sigma^2 I u_{ik} \\
&= \sigma^2 (u_{ij} \cdot u_{ik})
\end{aligned} \tag{50}$$

Considering that the linear terms dominate in the deviation expression, we can approximate:

$$\mathrm{Cov}(\Delta_{ij}, \Delta_{ik}) \approx \mathrm{Cov}(Z_{ij,u}, Z_{ik,u}) = \sigma^2 (u_{ij} \cdot u_{ik}) \tag{51}$$

The complete $3 \times 3$ covariance matrix for triangle edge deviations is:

$$\Sigma = \begin{bmatrix} \mathrm{Var}(\Delta_{ij}) & \mathrm{Cov}(\Delta_{ij}, \Delta_{ik}) & \mathrm{Cov}(\Delta_{ij}, \Delta_{jk}) \\ \mathrm{Cov}(\Delta_{ik}, \Delta_{ij}) & \mathrm{Var}(\Delta_{ik}) & \mathrm{Cov}(\Delta_{ik}, \Delta_{jk}) \\ \mathrm{Cov}(\Delta_{jk}, \Delta_{ij}) & \mathrm{Cov}(\Delta_{jk}, \Delta_{ik}) & \mathrm{Var}(\Delta_{jk}) \end{bmatrix} \tag{52}$$

Substituting the specific expressions:

$$\Sigma = \begin{bmatrix} 2\sigma^2 + \frac{3\sigma^4}{d_{ij}^2} & \sigma^2(u_{ij} \cdot u_{ik}) & \sigma^2(u_{ij} \cdot u_{jk}) \\ \sigma^2(u_{ik} \cdot u_{ij}) & 2\sigma^2 + \frac{3\sigma^4}{d_{ik}^2} & \sigma^2(u_{ik} \cdot u_{jk}) \\ \sigma^2(u_{jk} \cdot u_{ij}) & \sigma^2(u_{jk} \cdot u_{ik}) & 2\sigma^2 + \frac{3\sigma^4}{d_{jk}^2} \end{bmatrix} \tag{53}$$

The deviation vector of the triangle relation matrix is defined as:

$$\Delta D_{ijk} = D'_{ijk} - D_{ijk} = [\Delta_{ij}, \Delta_{ik}, \Delta_{jk}] \tag{54}$$

This vector follows a multivariate normal distribution:

$$\Delta D_{ijk} \sim \mathcal{N}(\mu, \Sigma) \tag{55}$$

with mean vector:

$$\mu = [\mathbb{E}[\Delta_{ij}], \mathbb{E}[\Delta_{ik}], \mathbb{E}[\Delta_{jk}]] = \left[ \frac{3\sigma^2}{d_{ij}}, \frac{3\sigma^2}{d_{ik}}, \frac{3\sigma^2}{d_{jk}} \right] \tag{56}$$

and covariance matrix $\Sigma$ as defined previously.

The Frobenius norm of the deviation matrix is:

$$\|\Delta D_{ijk}\|_F = \sqrt{\Delta_{ij}^2 + \Delta_{ik}^2 + \Delta_{jk}^2} \tag{57}$$

The squared norm follows a non-central chi-squared distribution with 3 degrees of freedom and non-centrality parameter:

$$\begin{aligned}
\lambda &= \|\mu\|^2 \\
&= \left(\frac{3\sigma^2}{d_{ij}}\right)^2 + \left(\frac{3\sigma^2}{d_{ik}}\right)^2 + \left(\frac{3\sigma^2}{d_{jk}}\right)^2 \\
&= 9\sigma^4 \left(\frac{1}{d_{ij}^2} + \frac{1}{d_{ik}^2} + \frac{1}{d_{jk}^2}\right) \\
&= 9\sigma^4 \sum_{(a,b) \in \{(i,j),(i,k),(j,k)\}} \frac{1}{d_{ab}^2}
\end{aligned} \tag{58}$$

For incorrect point correspondences, at least one point is mismatched or belongs to a different rigid body. In this case, the deviation includes both random noise and systematic geometric error:

$$\Delta D_{ijk} \sim \mathcal{N}(\mu + \delta_{\text{sys}}, \Sigma) \tag{59}$$

where $\delta_{\text{sys}} = [\delta_{ij}, \delta_{ik}, \delta_{jk}]$ represents the systematic error vector that typically satisfies $\|\delta_{\text{sys}}\| \gg \sigma$.

Given a threshold $\epsilon$, we define:

$$p_1 = \mathbb{P}(\|\Delta D_{ijk}\|_F < \epsilon \mid \text{correct correspondence}) \tag{60}$$

$$p_2 = \mathbb{P}(\|\Delta D_{ijk}\|_F < \epsilon \mid \text{incorrect correspondence}) \tag{61}$$

Under conditions of sufficient point cloud density, relatively small noise level $\sigma$, and significant systematic error $\|\delta_{\text{sys}}\| \gg \sigma$, we can establish that $p_1 > p_2$.

For correct correspondences, $\|\Delta D_{ijk}\|_F^2$ follows a non-central chi-squared distribution with non-centrality parameter $\lambda_1 = \|\mu\|^2$.

For incorrect correspondences, $\|\Delta D_{ijk}\|_F^2$ follows a non-central chi-squared distribution with non-centrality parameter $\lambda_2 = \|\mu + \delta_{\text{sys}}\|^2$.

Since $\|\delta_{\text{sys}}\| \gg \|\mu\|$, we have $\lambda_2 \gg \lambda_1$. The cumulative distribution function of a non-central chi-squared distribution decreases with increasing non-centrality parameter for a fixed threshold. Therefore:

$$p_1 = \mathbb{P}(\|\Delta D_{ijk}\|_F < \epsilon \mid \text{correct}) \gg \mathbb{P}(\|\Delta D_{ijk}\|_F < \epsilon \mid \text{incorrect}) = p_2 \tag{62}$$

For a $k$-truss, where each edge is supported by at least k-2 triangles, the probability of correctly identifying a correspondence increases exponentially with k, while the probability of incorrectly accepting a false correspondence decreases exponentially. Assuming approximate independence between triangles supporting an edge, the probability of correctly identifying a correspondence with m = k-2 supporting triangles is:

$$P(\text{correct} \mid m \text{ triangles}) \approx 1 - (1 - p_1)^m \tag{63}$$

Similarly, the probability of incorrectly accepting a false correspondence with m supporting triangles is:

$$P(\text{incorrect accepted} \mid m \text{ triangles}) \approx (p_2)^m \tag{64}$$

Since $p_1 > p_2$, as m increases (higher $k$-truss order), the discrimination power between correct and incorrect correspondences increases substantially. This provides the theoretical foundation for why triangle-supported $k$-truss offer exceptional robustness in point cloud registration, particularly in challenging scenarios with noise, outliers, and partial visibility.

## A.3 Computational Complexity Analysis

In this section, we provide a systematic comparison of the computational complexity for four representative dense subgraph algorithms commonly used in graph-based correspondence selection: maximum clique [32], maximal clique [48], $k$-truss (our method), and k-core[35]. As shown in Table 5, this analysis highlights the superior efficiency of $k$-truss for large-scale point cloud registration.

**maximum clique.**  The maximum clique problem aims to find the largest fully connected subgraph within a given graph. This is a classic NP-complete problem. The number of possible cliques grows exponentially with the number of vertices, making exact computation intractable for large graphs. The time complexity is exponential with respect to the number of nodes, and thus maximum clique algorithms are unsuitable for practical large-scale applications.

**maximal clique.**  A maximal clique is a clique that cannot be extended by including any adjacent vertex; it is not necessarily the largest clique, but it is maximal with respect to set inclusion. Enumerating all maximal cliques in a graph is also computationally demanding, as the number of such cliques can still be exponential in the worst case. As a result, MAC-based methods are robust in theory but suffer from high computational cost and limited scalability.

**$k$-truss (our method).**  $k$-truss decomposition efficiently finds a subgraph in which every edge is contained in at least $k - 2$ triangles. The main computational steps are triangle enumeration and iterative edge removal based on triangle support. The overall time complexity is polynomial, typically $O(m^{1.5})$ where $m$ is the number of edges. This makes $k$-truss much more efficient and scalable than clique-based methods, while still leveraging higher-order geometric consistency (triangles) for robust correspondence selection.

**k-core.**  k-core decomposition identifies the largest subgraph in which every node has at least degree k. It can be computed with a simple iterative node removal process in linear time, $O(m)$. This method is extremely efficient and suitable for very large graphs. However, k-core only considers node degree and ignores triangle or higher-order structures, which can limit its robustness when facing high outlier rates or complex geometric scenarios.

Table 5: Time complexity comparison of dense subgraph algorithms.

| Method | Description | Time Complexity | Structural Strength |
|---|---|---|---|
| maximum clique | Largest fully-connected subgraph | Exponential | Strongest, but intractable |
| maximal clique (MAC) | Maximal fully-connected subgraph | Exponential | Strong, but costly |
| $k$-truss | Each edge in $\geq k-2$ triangles | Polynomial ($O(m^{1.5})$) | Strong (triangle-based) |
| k-core | Each node degree $\geq k$ | Linear ($O(m)$) | Moderate (degree-based) |

Both maximum clique and maximal clique approaches impose strict connectivity constraints but have exponential time complexity, making them impractical for large-scale or real-time applications. k-core is extremely efficient but provides only weak structural guarantees. In contrast, $k$-truss decomposition achieves a favorable balance: it maintains polynomial computational complexity, making it feasible for large-scale graphs, while its triangle-based strpucture ensures robust inlier selection. This advantage explains the superior efficiency and effectiveness of $k$-truss in our framework for large-scale point cloud registration. Moreover, parallel $k$-truss decomposition algorithms can further improve the speed of $k$-truss extraction, making it even more suitable for large-scale applications.

## A.4 Pseudocode for Our Algorithm

The following pseudocode outlines the complete pipeline of **PointTruss**, our robust 3D point cloud registration framework. The overall method consists of four modular stages: (1) consensus voting-based sampling, (2) compatibility graph construction and $k$-truss decomposition, (3) cluster-wise transformation estimation, and (4) spatial distribution score-based selection. Each stage is further detailed below with interleaved explanation and modular pseudocode.

---
**Algorithm 1** PointTruss
---
1: **Input:** Source point cloud $\mathbf{P}$, target point cloud $\mathbf{Q}$, initial correspondences $\mathcal{C}$, parameters: noise std $\sigma$, inlier threshold $\tau$, sampling ratio $\beta$, $k$-truss parameter k, consensus threshold $\tau_c$, compatibility threshold $\tau_{\text{comp}}$, NMS radius $r_{\text{nms}}$
2: **Output:** Rigid transformation $(\mathbf{R}^*, \mathbf{t}^*)$
3: Apply Algorithm 2 to $\mathcal{C}$ with $\tau_c$, $r_{\text{nms}}$, $\beta$ to obtain $\mathcal{I}_{\text{sampled}}$
4: Let $\mathcal{C}_{\text{sampled}} = \{\mathcal{C}[i] : i \in \mathcal{I}_{\text{sampled}}\}$
5: Construct compatibility graph $G$ among $\mathcal{C}_{\text{sampled}}$; add edge between $c_i$ and $c_j$ if $S_{\text{comp}}(c_i, c_j) > \tau_{\text{comp}}$, using noise parameter $\sigma$
6: Compute adjacency matrix $\mathbf{A}$ of $G$
7: Apply Algorithm 3 to $\mathbf{A}$ with k to get robust clusters $\{\mathcal{N}_m\}$
8: **for** each cluster $\mathcal{N}_m$ **do**
9:     Compute node centrality in cluster to obtain weights $w_i$
10:    Estimate candidate transformation $(\mathbf{R}_m, \mathbf{t}_m)$ using centrality-weighted SVD
11: **end for**
12: Apply Algorithm 4 to all candidate transformations $\{(\mathbf{R}_m, \mathbf{t}_m)\}$ to obtain optimal $(\mathbf{R}^*, \mathbf{t}^*)$
13: **return** $(\mathbf{R}^*, \mathbf{t}^*)$
---

### Step 1: Consensus Voting-based Low-scale Sampling.

Given initial correspondences, we first select a subset of high-quality matches via consensus voting and non-maximum suppression (NMS). This improves the precision of the subsequent graph construction.

---
**Algorithm 2** Consensus Voting-based Low-scale Sampling
---
1: **Input:** Correspondences $\mathcal{C} = \{(\mathbf{p}_i^s, \mathbf{p}_i^t)\}_{i=1}^N$, compatibility scores $S_{\text{comp}}(\cdot, \cdot)$, consistency threshold $\tau_c$, NMS radius $r_{\text{nms}}$, sampling ratio $\beta$
2: **Output:** Sampled correspondence indices $\mathcal{I}_{\text{sampled}}$
3: **for** each correspondence $i = 1$ to $N$ **do**
4:     Compute consensus score $S_i$ using $S_{\text{comp}}$ and threshold $\tau_c$ (please see Eq. (3))
5: **end for**
6: **for** each correspondence $i = 1$ to $N$ **do**
7:     Apply non-maximum suppression to $S_i$ using NMS radius $r_{\text{nms}}$ (please see Eq. (5))
8:     Compute final score $S_i^{\text{final}}$ (please see Eq. (6))
9: **end for**
10: Set $K = \lfloor \beta \cdot N \rfloor$
11: Select indices $\mathcal{I}_{\text{sampled}}$ of the top-$K$ correspondences with the highest $S_i^{\text{final}}$
12: **return** $\mathcal{I}_{\text{sampled}}$
---

### Step 2: Compatibility Graph Construction and K-Truss Decomposition.

A compatibility graph is built over the sampled correspondences, where edges represent geometric consistency. We then extract structurally robust clusters using k-truss decomposition.

### Step 3: Cluster-wise Transformation Estimation.

For each cluster, we estimate a candidate rigid transformation using centrality-weighted SVD. All candidate transformations are subsequently evaluated in the next step.

### Step 4: Spatial Distribution Score (SDS) Based Selection.

We score each candidate using SDS, which measures both the alignment quality and spatial spread of inliers, and select the best one for output.

### A.5 Hyper-parameter selection

We set the inlier threshold $\tau$ to 0.1 for the 3DMatch and 3DLoMatch datasets. For the KITTI dataset, $\tau$ is set to 0.6. The sampling ratio $\beta$ ranges from 0.1 to 0.5. The k-truss parameter k is chosen between 3 and 10. The consensus threshold $\tau_c$ is set to 0.9 by default. The compatibility threshold

---

**Algorithm 3** K-Truss Decomposition for Enhanced Structure Detection

---

1: **Input:** Adjacency matrix $\mathbf{A}$ of the compatibility graph among sampled correspondences, truss parameter k
2: **Output:** Robust clusters of correspondences $\{\mathcal{N}_m\}$
3: Compute triangle support matrix $\mathbf{T}$ using $\mathbf{A}$
4: Identify valid edges $\mathbf{E}_{\text{valid}}$ where triangle support $\geq (k-2)$
5: **for** each vertex $i$ **do**
6:     Extract neighborhood $\mathcal{N}_i$ connected by valid edges
7:     **if** $|\mathcal{N}_i| \geq k$ **then**
8:         Add $\mathcal{N}_i$ to the set of robust clusters $\{\mathcal{N}_m\}$
9:     **end if**
10: **end for**
11: **return** $\{\mathcal{N}_m\}$

---

---

**Algorithm 4** Spatial Distribution Score (SDS) Based Transformation Selection

---

1: **Input:** Source points $\mathbf{P} = \{\mathbf{p}_i^s\}_{i=1}^N$, target points $\mathbf{Q} = \{\mathbf{p}_i^t\}_{i=1}^N$, candidate transformations $\{(\mathbf{R}_m, \mathbf{t}_m)\}_{m=1}^M$, inlier threshold $\tau$
2: **Output:** Optimal transformation $(\mathbf{R}^*, \mathbf{t}^*)$
3: **for** each candidate $(\mathbf{R}_m, \mathbf{t}_m)$ **do**
4:     Transform source points using $(\mathbf{R}_m, \mathbf{t}_m)$
5:     Identify inlier set $\mathcal{I}_m$ using threshold $\tau$
6:     **if** $|\mathcal{I}_m| < 10$ **then**
7:         Assign SDS score $= 0$ for this candidate
8:     **else**
9:         Compute inlier ratio $\rho_{\text{inlier}}$
10:         Compute spatial coverage ratio $\rho_{\text{coverage}}$
11:         Compute inlier alignment error term $\rho_{\text{error}}$
12:         Compute SDS score (please see Eq. (14–17))
13:     **end if**
14: **end for**
15: Select $(\mathbf{R}^*, \mathbf{t}^*)$ with the highest SDS score
16: **return** $(\mathbf{R}^*, \mathbf{t}^*)$

---

$\tau_{\text{comp}}$ is adjusted according to the noise standard deviation $\sigma$. The NMS radius $r_{\text{nms}}$ is typically set to 0.1. All hyperparameters are determined based on empirical validation.

### A.6 Ablation study of each component

We conducted systematic ablation studies on the 3DMatch and 3DLoMatch datasets to analyze each component of our algorithm. The MAC (Maximal Clique) method was introduced for comparison, ensuring a comprehensive evaluation of our proposed modules. As shown in Table 6, our method demonstrates the effectiveness of the k-truss for point cloud registration. This structure not only improves overall registration accuracy but also enhances robustness. In addition, the consensus voting-based low-scale sampling strategy and spatial distribution score each contribute positively in experiments. Results indicate that both strategies can serve as effective components for traditional registration methods, improving their performance in challenging scenarios. Overall, our study confirms the practical value and broad applicability of these modules in point cloud registration tasks.

### A.7 Limitations and broader impacts

We propose a novel point cloud registration method based on the k-truss in graph theory. This method uses triangles as core constraints and introduces a new perspective for point cloud registration. It fully exploits the advantages of higher-order structures in modeling spatial relationships. Our method performs well in both dense and sparse point cloud scenarios. It shows strong robustness and generalization, effectively resisting high ratios of outliers and noise. In addition, the k-truss decomposition module in our algorithm is highly extensible. It can be used as an independent

Table 6: Analysis experiments on 3DMatch and 3DLoMatch with FPFH and FCGF descriptors. **CV**: Consensus Voting-based Low-scale Sampling Strategy; **MAC**: Maximal Clique; **SDS**: Spatial Distribution Score.

| | CV | MAC | k-truss | SDS | inlier | $RR_{3DMatch}(\%)$ | $RR_{3DLoMatch}(\%)$ |
|---|---|---|---|---|---|---|---|
| **FPFH** | | | | | | | |
| 1) | | ✓ | | | ✓ | 83.67 | 37.10 |
| 2) | ✓ | ✓ | | | ✓ | 83.80 | 38.85 |
| 3) | | ✓ | | ✓ | | 83.73 | 39.02 |
| 4) | ✓ | ✓ | | ✓ | | 84.20 | 38.91 |
| 5) | | | ✓ | | ✓ | 83.86 | 37.79 |
| 6) | ✓ | | ✓ | | ✓ | 84.20 | 39.98 |
| 7) | | | ✓ | ✓ | | 83.86 | 38.85 |
| 8) | ✓ | | ✓ | ✓ | | **84.70** | **43.96** |
| **FCGF** | | | | | | | |
| 1) | | ✓ | | | ✓ | 91.68 | 57.44 |
| 2) | ✓ | ✓ | | | ✓ | 93.59 | 59.46 |
| 3) | | ✓ | | ✓ | | 93.53 | 59.01 |
| 4) | ✓ | ✓ | | ✓ | | 93.72 | 59.96 |
| 5) | | | ✓ | | ✓ | 93.40 | 58.00 |
| 6) | ✓ | | ✓ | | ✓ | 93.72 | 59.63 |
| 7) | | | ✓ | ✓ | | 93.66 | 59.40 |
| 8) | ✓ | | ✓ | ✓ | | **93.84** | **61.64** |

component and flexibly integrated into other registration algorithms or point cloud processing frameworks, further enhancing overall system performance.

In terms of applications, this method is particularly suitable for scenarios requiring high precision and stability in point cloud registration, such as autonomous driving. For example, in perception and localization tasks for autonomous vehicles, reliable point cloud registration is crucial for environment understanding and high-precision map construction. Nevertheless, our method still has room for improvement in the adaptive selection of the k value and the screening of the optimal k-truss subgraph. At present, how to automatically determine the best k value according to different data characteristics, and how to efficiently select representative k-truss substructures, are the main directions for our future research.

In the future, we will further explore the potential of higher-order topological structures, such as quadrilaterals, in point cloud registration. This will improve the expressive power of the algorithm in complex scenarios. We also plan to leverage high-performance parallel computing frameworks, such as PyTorch, to parallelize the k-truss decomposition process. This will enable real-time processing of large-scale point cloud data. Through these improvements, we aim to promote the application of graph-based point cloud registration algorithms in real engineering scenarios and advance development in related fields.

### A.8 Scalability of our algorithm

Our experimental results demonstrate that the proposed algorithm achieves state-of-the-art performance in both accuracy and robustness, while also exhibiting excellent efficiency. With further optimization, such as leveraging PyTorch's parallel computation capabilities, the speed of our method can be further improved. Currently, the k-truss decomposition accounts for most of the runtime. Several studies have optimized parallel k-truss decomposition, and these techniques can also be applied to our method. In addition, our method can serve as a flexible module that integrates with conventional or deep learning-based descriptors. The proposed Spatial Distribution Score can also be adopted by other methods to further enhance overall accuracy.

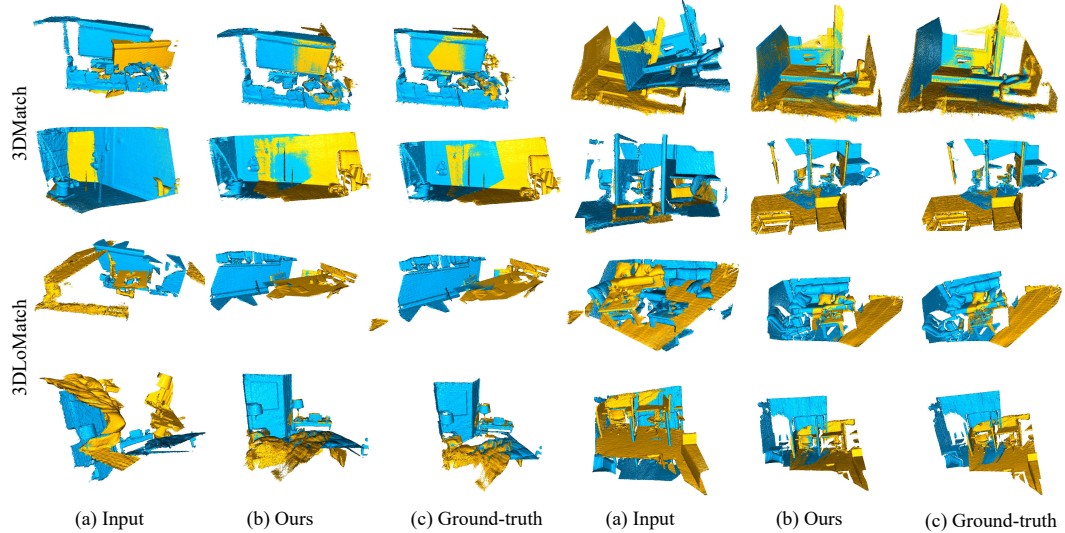

|  | 3DMatch |  |  |  |  |  |
|---|---|---|---|---|---|---|
| | (a) Input | (b) Ours | (c) Ground-truth | (a) Input | (b) Ours | (c) Ground-truth |

Figure 6: **Visualizations of registration results on the 3DMatch and 3DLoMatch datasets.** The first two rows show examples from 3DMatch, and the last two rows show examples from 3DLoMatch. In each group, yellow and blue point clouds represent the source and target, respectively. From left to right: (a) input point cloud pairs, (b) results of our method, and (c) ground-truth alignment.

## A.9 Datasets

All datasets used in this work are publicly available. The Bunny model from the Stanford 3D Scanning Repository was acquired using a Cyberware 3030 MS scanner and is restricted to non-commercial use. The KITTI dataset is published under the NonCommercial-ShareAlike 3.0 License and contains 11 sequences captured by a Velodyne HDL-64 3D LiDAR scanner in outdoor driving scenarios. Following the protocol in [5, 48], we use sequences 8–10 for testing. Additionally, we provide the 3DMatch dataset and its corresponding license information, as shown in Table 7, where 3DLoMatch is a subset of 3DMatch.

Table 7: Source datasets for 3DMatch and their corresponding licenses.

| Datasets | License |
|---|---|
| SUN3D [38] | CC BY-NC-SA 4.0 |
| 7-Scenes [33] | Non-commercial use only |
| RGB-D Scenes v.2 [22] | (License not stated) |
| Analysis-by-Synthesis [36] | CC BY-NC-SA 4.0 |
| BundleFusion [11] | CC BY-NC-SA 4.0 |
| Halber et al. [16] | CC BY-NC-SA 4.0 |

## A.10 Visualization of Registration Results

We present visual registration results on the 3DMatch and 3DLoMatch datasets in Fig. 6. The yellow and blue point clouds represent the source and target, respectively. The first column shows the input point clouds, and the second column displays the point clouds aligned using the ground-truth transformation. Even on the 3DLoMatch dataset with low overlap, our method clearly extracts the key structure and achieves accurate alignment. We also show registration results on the KITTI dataset in Fig. 7. The input source and target point clouds are in different poses. After applying our estimated transformation, the source point cloud is successfully aligned with the target. The result is almost identical to that of the ground-truth transformation.

| Input | Ours | Ground-truth |
|:-:|:-:|:-:|

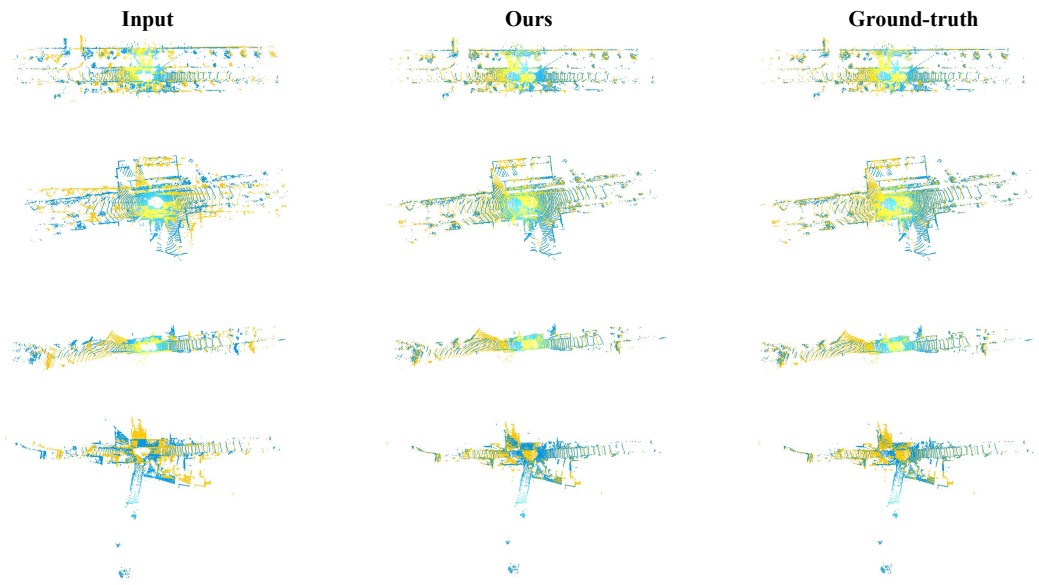

Figure 7: **Visualizations of registration results on the KITTI dataset.** From left to right: input point cloud pairs, registration results using our method, and ground-truth alignment.

### A.11 Ablation Study on Graph Sampling and K-value

We conduct comprehensive ablations to evaluate how the sampling ratio $\beta$ influences accuracy and efficiency, as shown in Table 8.

Table 8: Impact of sampling ratio $\beta$ on 3DMatch (FCGF).

| Sampling Ratio $\beta$ | Registration Recall | Runtime (s) | Speedup |
|:--|:-:|:-:|:-:|
| 0.1 | 93.10% | 0.12 | 8.3× |
| 0.2 | 93.53% | 0.16 | 6.3× |
| 0.3 | 93.84% | 0.20 | 5.0× |
| 0.4 | 93.78% | 0.25 | 4.0× |
| 0.5 | 93.41% | 0.31 | 3.2× |

Retaining only 10% of correspondences via consensus voting preserves 93.10% registration success while yielding an 8.3× speedup, indicating that voting preferentially keeps geometrically consistent correspondences. Higher sampling ratios can be slightly worse than $\beta$=0.3 because they retain more erroneous correspondences that adversely affect the downstream k-truss decomposition.

Initial graph: 5,020 nodes → after voting: 502 nodes → adjacency: 502×502 with 43,571 edges.

Table 9: K-Truss decomposition statistics with 10% sampling.

| k-value | Subgraphs | Node Range | Avg. Nodes |
|:--|:-:|:-:|:-:|
| 3 | 502 | [3–322] | 173.6 |
| 4 | 502 | [4–322] | 173.6 |
| 5 | 499 | [5–322] | 174.5 |
| 6 | 498 | [6–322] | 174.7 |
| 7 | 495 | [7–320] | 175.5 |
| 8 | 493 | [8–321] | 176.0 |
| 9 | 492 | [9–320] | 176.1 |
| 10 | 490 | [10–320] | 176.4 |

As shown in Table 9, even with aggressive 10% sampling, subgraphs remain large (average 170+ nodes), due to: (i) high-quality input after the voting pre-filter that enforces strong geometric consistency and high connectivity density; and (ii) natural clustering of correct correspondences, which form richly supported triangle structures satisfying k-truss constraints.

**Sensitivity to $k$ and Multi-$k$**  The sensitivity of registration recall to k is shown in Table 10.

Table 10: Sensitivity of registration recall to $k$ on 3DMatch with FCGF.

| k-value | Registration Recall |
|---|---|
| 3 | 93.15% |
| 5 | 93.21% |
| 7 | 93.59% |
| 10 | 93.40% |
| Multi-$k$ (3–10) | 93.84% |

Performance is weakly sensitive to $k$ because stricter $k$-truss subgraphs are nested subsets of those at smaller $k$, forming a natural hierarchy. The multi-$k$ strategy leverages this property. In practice, we select the transformation from the subgraph with the highest spatial distribution score; a global re-estimation using all inliers is unnecessary, as the selected subgraph already yields robust alignment.

