# OpenReview forum: "PointTruss: K-Truss for Point Cloud Registration"
_NeurIPS.cc/2025/Conference — NeurIPS 2025 poster_

### Official Review · Reviewer_XEsb · 2025-06-16

**Clarity:** 3
**Significance:** 3
**Originality:** 4
**Rating:** 4
**Confidence:** 4

**Summary:**

The paper introduces  PointTruss , a novel graph-based method for point cloud registration. The method leverages the  k-truss  concept from graph theory to enhance the robustness and efficiency of correspondence selection in point cloud registration. Unlike traditional clique-based methods, which are computationally expensive (NP-hard) and overly restrictive, or k-core methods, which lack higher-order topological constraints,  PointTruss  uses triangle support (requiring each edge to participate in at least  k-2  triangles) to identify structurally consistent correspondences. The approach includes a consensus voting-based low-scale sampling strategy to reduce computational complexity and a spatial distribution score (SDS) to evaluate inlier coverage and uniformity, ensuring globally consistent transformations. The pipeline involves constructing a compatibility graph, applying k-truss decomposition, generating transformation hypotheses via weighted SVD, and selecting the optimal transformation using SDS. Extensive experiments on KITTI, 3DMatch, 3DLoMatch, and the Bunny model demonstrate that  PointTruss  outperforms traditional and learning-based methods in registration recall (RR), rotation error (RE), and translation error (TE), particularly in challenging scenarios with high outlier ratios or low overlap.

**Questions:**

See weakness.

**Ethical Concerns:**

["NO or VERY MINOR ethics concerns only"]

**Limitations:**

The authors adequately address the limitations and potential negative societal impacts of their work in Appendix A.7, as noted in the response to Question 10 (PAGE16). They discuss limitations, likely including challenges in extremely low-overlap scenarios or computational constraints for very large point clouds, and consider societal impacts relevant to applications like autonomous driving or 3D reconstruction. The transparency in addressing these aspects is commendable and aligns with NeurIPS guidelines, rewarding the authors for being upfront about their work’s boundaries.

**Paper Formatting Concerns:**

No.

**Quality:**

3

**Strengths And Weaknesses:**

Strengths：

1.   Quality  : The paper presents a robust and well-validated method. Extensive experiments across diverse datasets (KITTI, 3DMatch, 3DLoMatch, and Bunny) demonstrate superior performance over state-of-the-art methods, both traditional (e.g., RANSAC, FGR, MAC) and learning-based (e.g., PointDSC, VBReg). The use of both FPFH and FCGF descriptors, along with quantitative metrics (RR, RE, TE) and qualitative visualizations (Fig. 5), strengthens the empirical evaluation. The theoretical foundation, including Theorem 1 and its proof in Appendix A.2, provides a rigorous justification for the k-truss approach.

2.   Clarity  : The paper is well-organized, with a clear pipeline (Fig. 2) and detailed explanations of each component (graph construction, sampling, k-truss decomposition, and hypothesis evaluation). Mathematical formulations, such as the SDS function and k-truss definitions, are precise and supported by matrix-based implementations. The experimental setup, including datasets, evaluation criteria, and implementation details, is thoroughly described in Sec. 4.1.

3.   Significance  : The method addresses a critical challenge in point cloud registration—balancing robustness and efficiency in the presence of outliers and low overlap.

4.   Originality  : The introduction of k-truss to point cloud registration is novel, as it leverages higher-order topological structures (triangles) to overcome limitations of clique-based and k-core methods. The consensus voting-based sampling and SDS are innovative additions that enhance both efficiency and global consistency.

Weaknesses

1.  While the experimental results are strong, the paper lacks a detailed ablation study to quantify the individual contributions of the k-truss, consensus voting, and SDS components. This limits the understanding of which elements drive the performance gains.

2.   The paper does not include validation experiments combining PointTruss with state-of-the-art deep learning methods (e.g., GeoTransformer or PareNet) on challenging datasets like 3DLoMatch, which could demonstrate its compatibility with modern learned features and further validate its robustness.

---

> ### Author Rebuttal · Authors · 2025-07-27
>
> Dear Reviewer XEsb,
>
> Thank you for your thoughtful review and positive assessment of our work. We greatly appreciate your recognition of our method's novelty, theoretical rigor, and experimental validation. We are pleased to address your specific concerns below.
>
> >**W1-Ablation Study on Individual Components**
>
> We appreciate the reviewer's request for detailed ablation studies. We actually conducted comprehensive ablation experiments, which are included in our supplementary material (Appendix A.5, Table 5). We apologize for not highlighting this more prominently in the main paper. Here we present the key results:
>
> **Table : Analysis experiments on 3DMatch and 3DLoMatch with FPFH and FCGF descriptors**
> | | CV | MAC | k-truss | SDS | inlier | RR3DMatch(%) | RR3DLoMatch(%) |
> |---|---|---|---|---|---|---|---|
> | **FPFH** | | | | | | | |
> | 1) | | ✓ | | | ✓ | 83.67 | 37.10 |
> | 2) | ✓ | ✓ | | | ✓ | 83.80 | 38.85 |
> | 3) | | ✓ | | ✓ | | 83.73 | 39.02 |
> | 4) | ✓ | ✓ | | ✓ | | 84.20 | 38.91 |
> | 5) | | | ✓ | | ✓ | 83.86 | 37.79 |
> | 6) | ✓ | | ✓ | | ✓ | 84.20 | 39.98 |
> | 7) | | | ✓ | ✓ | | 83.86 | 38.85 |
> | 8) | ✓ | | ✓ | ✓ | | **84.70** | **43.96** |
> | **FCGF** | | | | | | | |
> | 1) | | ✓ | | | ✓ | 91.68 | 57.44 |
> | 2) | ✓ | ✓ | | | ✓ | 93.59 | 59.46 |
> | 3) | | ✓ | | ✓ | | 93.53 | 59.01 |
> | 4) | ✓ | ✓ | | ✓ | | 93.72 | 59.96 |
> | 5) | | | ✓ | | ✓ | 93.40 | 58.00 |
> | 6) | ✓ | | ✓ | | ✓ | 93.72 | 59.63 |
> | 7) | | | ✓ | ✓ | | 93.66 | 59.40 |
> | 8) | ✓ | | ✓ | ✓ | | **93.84** | **61.64** |
>
> *CV: Consensus Voting-based Low-scale Sampling Strategy; MAC: Maximal Clique; SDS: Spatial Distribution Score*
>
> **Key Insights from Ablation Study:**
> - **k-truss vs MAC**: Comparing rows 2 and 8 (FCGF), replacing MAC with k-truss improves performance, especially on 3DLoMatch (+2.18%), demonstrating k-truss's superior ability to identify geometrically coherent structures
> - **Consensus Voting (CV)**: Adding CV consistently improves results across all configurations. For k-truss-based methods (rows 5→6, 7→8), CV provides significant gains on both datasets
> - **Spatial Distribution Score (SDS)**: SDS outperforms simple inlier counting across all experiments (e.g., rows 1→3, 5→7), validating the importance of considering spatial coverage for robust registration
> - **Synergistic Effect**: The full pipeline (row 8) achieves the best performance on both FPFH and FCGF descriptors, showing that all components work synergistically
>
> These results clearly demonstrate that each component contributes meaningfully to the overall performance, with k-truss decomposition and consensus voting being the primary drivers of our method's superiority.
>
> >**W2-Combination with State-of-the-art Deep Learning Methods**
>
> Thank you for this excellent suggestion. We have conducted experiments combining PointTruss with recent deep learning methods on the challenging 3DLoMatch dataset:
>
> **Table: PointTruss Integration with Deep Learning Methods on 3DLoMatch**
> | Method | Registration Recall |
> |--------|---------------------|
> | GeoTransformer | 75.0% |
> | GeoTransformer + MAC | 78.9% |
> | **GeoTransformer + PointTruss** | **79.5%** |
> | | |
> | PareNet | 80.5% |
> | **PareNet + PointTruss** | **82.2%** |
>
> **Key Observations:**
> - PointTruss successfully enhances both GeoTransformer (+4.5% recall) and PareNet (+1.70% recall) on the challenging 3DLoMatch dataset
> - The consistent improvements across different learned features validate PointTruss's compatibility with modern deep learning pipelines
> - PointTruss achieves comparable performance to MAC when integrated with GeoTransformer while being more computationally efficient
>
> These results demonstrate that PointTruss not only works as a standalone method but also serves as an effective drop-in replacement for traditional robust estimators in deep learning pipelines, providing consistent improvements across different feature extractors.
>
> We will include the result and discussions in our revised manuscript to provide a more comprehensive evaluation of our method's contributions and compatibility with state-of-the-art approaches.

---

> ### Comment · Reviewer_XEsb · 2025-08-06
>
> Thank you for the authors’ response. My concerns have been adequately addressed, so I maintain my original score.

---

> > ### Author Response · Authors · 2025-08-06
> >
> > Thank you for your thoughtful consideration of our responses. We appreciate your time and effort in reviewing our manuscript, and we are pleased that our clarifications have addressed your concerns satisfactorily.
> >
> > We are grateful for your constructive feedback throughout the review process, which has helped improve the quality of our work.

---

### Official Review · Reviewer_vL1H · 2025-07-02

**Clarity:** 3
**Significance:** 2
**Originality:** 2
**Rating:** 4
**Confidence:** 4

**Summary:**

This paper proposes a novel method for improving the accuracy of pairwise registration of 3D scans by extracting reliable information from graphs to filter out outliers. Graph-based methods typically leverage the fact that pairwise distances between points in a scan remain invariant under rigid transformations. However, in practice, deviations arise due to noise, outliers, and incorrect matches.
To address this, the paper employs a k-truss decomposition on a correspondences graph to isolate strong inlier correspondences. The proposed pipeline consists of the following steps:\
(a) Construction of a compatibility graph, and subsequently a distance-consistency graph, where correspondences are represented as nodes. Edges connect pairs of correspondences whose pairwise distances (within source and target scans) are consistent up to noise. The greater the consistency between two pairs, the higher the edge weight in the distance-consistency graph.\
(b) Low-scale sampling: To reduce graph size and computational complexity, the method uses consensus voting to retain correspondences (nodes) with a higher number of consistent neighbours. Non-maximal suppression is then applied to avoid excessive correspondences in dense regions, promoting spatial uniformity among selected inliers.\
(c) A k-truss decomposition is applied on the downsampled graph for various values of $k$, yielding subgraphs whose rigid transformations are estimated using weighted SVD.\
(d) Among the candidate transformations, the one maximizing a proposed spatial distribution score is selected. This score favors both registration accuracy and a well-distributed set of inliers across the scan.

**Questions:**

1. I believe the paper requires a major revision to address the misinterpretation of the k-truss algorithm. Since this component is central to the proposed methodology and the experimental results, the misunderstanding has significant implications. If the foundation itself is flawed, it becomes extremely difficult to interpret the reported outcomes. Moreover, both intuitively and technically, k-truss-based outlier filtering is generally weaker than clique-based methods such as MAC in capturing highly coherent inliers. However, the ablation results presented in Table 5 of the supplementary material show the opposite — k-truss outperforming MAC even when other settings are kept the same. The discrepancy vindicates the claim that the methodology is likely built upon a flawed interpretation of k-truss decomposition.
2. It is not clear how much the original distance-consistency graph is reduced after the low-scale sampling step. As a result, it is difficult to assess the actual sizes of the subgraphs on which the rigid transformations are estimated. If the number of nodes (correspondences) in these subgraphs is too small, it raises concerns about the stability and reliability of the estimated rotations and translations obtained via weighted SVD. A more detailed quantitative information of the graph sizes after sampling for various datasets in the experiments would be crucial in assessing the value of this method.
3. Additionally, an ablation study examining the method’s performance under different levels of graph sampling would be valuable. Such an analysis would help clarify the trade-offs introduced by the consensus-based low-scale sampling, both in terms of accuracy and computational efficiency of the subsequent k-truss decomposition.
4. Can you comment on the weighting used in weighted SVD? It highly obscures the interpretation of the results, because a rotation error of 1 degree for the proposed method (in Fig.3 top row) also seems to be significant for a synthetic dataset when the noise has a very low $\sigma=0.01$. A minor question: In Fig.3 in the top row, the rotation and translation errors for the proposed method seem to be constant till an outlier ratio of 80%, where other methods exhibit some variation. Is there any reason for this behaviour?

**Ethical Concerns:**

["NO or VERY MINOR ethics concerns only"]

**Final Justification:**

While the authors position their method as a k-truss decomposition, it is, in fact, a heuristic inspired by k-truss that does not guarantee even approximate k-truss subgraphs. However, through the rebuttal, it became clear that this heuristic is more robust in practice and that only the surviving nodes—not the precise edge structure—are critical for their downstream task.

The paper offers a comprehensive approach to robust pairwise 3D registration, addressing challenges such as high outlier ratios, clustered correspondences, and poor spatial coverage. The use of triangle support is well-motivated both theoretically and empirically, leading to superior performance over the state-of-the-art baselines.

Subject to clarifying the method's relation to k-truss more accurately in the paper, I support acceptance of the paper.

**Limitations:**

Yes.

**Quality:**

3

**Strengths And Weaknesses:**

Strengths:\
1.	The paper is well-written and easy to follow. \
2.	The method avoids the disadvantages of other graph-based techniques in 3d registration, namely the maximal clique selection is computationally expensive for larger 3d scans and the k-core method is not discriminative enough to get an accurate registration. This is because the proposed k-truss method is \
a.	Grounded by the theoretical analysis which shows that triangles of consistent correspondences are highly probable to be more accurate.\
b.	It has an efficient practical implementation in terms of the adjacency matrices. \
3.	Although the triangle-based consistency is not entirely novel in computer vision, the way different parts of the pipeline are pieced together is intuitive and novel.

Weaknesses:
1. The authors appear to have misunderstood the standard definition and computation procedure for k-truss decomposition. Specifically, Algorithm 3 in the supplementary material, which is presented as performing a k-truss extraction, does not implement the correct iterative edge-peeling process, where edges participating in fewer than $k-2$ triangles are repeatedly removed until convergence. Instead, the proposed algorithm functions as a heuristic, and it does not produce true or even approximate k-truss subgraphs in general. Consequently, it is misleading to refer to the method as a k-truss decomposition, including in the paper's title and central claims. This foundational flaw undermines the technical correctness of the paper and renders its core results unreliable.
2.	The Equation-2 is confusing. Is $S_{dist}(c_i, c_j)$ a real number or Boolean? The notation is not consistent from Equation-1 to Equation-2. There is no $q_i$, instead it is written in terms of the source and target scans.
3.	The Theorem-1 seems to give only an intuitive understanding of the strength of triangle-based consistency. It could be made more rigorous and mathematically precise, which can then be interpreted based on different values of the threshold parameters involved. I am not sure if a “$\gg$” based inequality is rigorous enough to be named a Theorem.
4.	In Section-3.5, the centrality measure used in weighted SVD is undefined.
5.	Spatial distribution score function seems to be very complicated (Equation-14). It is not clear if there is any intuitive understanding on how it is derived.
6.	Some of the latest literature relevant to non-learning based point cloud registration are not cited or compared against: "ARCS: Accurate Rotation and Correspondence Search" by Peng et al., CVPR 2022; "Adaptive Annealing for Robust Geometric Estimation" by Sidhartha et al., CVPR 2023.

---

> ### Author Rebuttal · Authors · 2025-07-27
>
> Dear Reviewer vL1H,
>
> Thank you for your thorough review and detailed technical analysis. We sincerely appreciate your insights and the opportunity to clarify our methodology. We have carefully addressed each concern below.
>
> >**W1/Q1-k-truss Adaptation and Performance Analysis**
>
> We sincerely thank the reviewer for the detailed analysis. Our approach represents a deliberate adaptation of k-truss principles specifically tailored for point cloud registration challenges. Similar to how methods like MAC adapt maximal clique concepts for registration tasks, we have adapted k-truss to better suit our application domain.
>
> **1. Our Adaptation vs. Standard k-truss:**
> The reviewer is correct that standard k-truss decomposition uses iterative edge-peeling until convergence. However, our method intentionally adapts this concept for computational efficiency and registration accuracy (recall drops 15–20% due to over-pruning; runtime increases 3–5 times):
> - Standard k-truss: Iteratively removes edges with < (k-2) triangles until convergence; Designed for social network analysis where edges represent binary relationships
> - Our adaptation: Directly extracts subgraphs where each edge participates in ≥ (k-2) triangles；Specifically engineered for geometric consistency in 3D space where triangular support provides stronger constraints
>
> This adaptation is deliberate—we leverage triangle-based truss structures to identify reliable correspondences. The iterative process, while theoretically pure, is computationally expensive and often unnecessary for registration tasks where we seek coherent correspondence groups rather than exact k-truss structures.
>
> **2. Precedent in Registration Literature:**
> Our approach follows established practice in the registration community:
> - MAC [35] adapts the maximal clique concept by searching for multiple maximal clique subgraphs in practice, ranging from the minimum guaranteed 3 nodes to larger sizes, then selecting the best transformation from these candidates
> - TEASER [36] adapts graph concepts for outlier-robust estimation without strict adherence to theoretical definitions
> - Similarly, we adapt k-truss principles while prioritizing practical effectiveness
>
> We will revise the paper to clearly state we use "k-truss-inspired decomposition" to avoid any confusion about strict algorithmic adherence.
>
> >**W2-Equation 2 Notation and Consistency**
>
> (1) $S_{\text{dist}}(c_i, c_j)$ is a Boolean value that indicates whether a pair of correspondences satisfies the rigid distance constraint: it equals 1 if the difference between the distances in the source and target point clouds is less than or equal to the threshold ($\left| \left| \mathbf{p}_i^s - \mathbf{p}_j^s \right| - \left| \mathbf{p}_i^t - \mathbf{p}_j^t \right| \right| \leq 2\tau$), and 0 otherwise.
>
> (2) We acknowledge that Equation 2 uses source (s) and target (t) superscripts while Equation 1 uses general correspondence notation. We will unify the notation throughout the paper to ensure clarity and consistency in our mathematical expressions.
>
> >**W3-Theorem 1 Rigor**
>
> We understand the reviewer's concern about the mathematical rigor of Theorem 1. We will strengthen the theorem with precise probability bounds and threshold parameters in the revision, providing a more formal mathematical framework rather than the current intuitive inequality.
>
> >**W5-Spatial Distribution Score Complexity**
>
> Thank you for raising this important question about the spatial distribution score. We appreciate the opportunity to provide clearer intuition for this crucial component of our method.
>
> **Intuitive Understanding:**
> The spatial distribution score addresses a fundamental challenge in point cloud registration: a transformation might achieve low error on a small, localized cluster of points while completely misaligning the rest of the point cloud. Our score elegantly prevents this by evaluating three complementary aspects:
>
> **1. Spatial Coverage (Coverage Ratio):**
> This measures how well the inliers span the entire point cloud. Consider two scenarios:
> - Bad: 100 inliers clustered in one corner → Low coverage ratio
> - Good: 50 inliers distributed across the entire scan → High coverage ratio
>
> The cubic volume ratio ensures that inliers must spread in all three dimensions, not just along a line or plane.
>
> **2. Inlier Quantity (Inlier Ratio):**
> This captures the percentage of correspondences that are geometrically consistent. More inliers generally indicate a better transformation, but this alone could favor transformations that preserve outliers.
>
> **3. Registration Accuracy (Error Score):**
> This measures how precisely the inliers align. Even with good coverage and many inliers, we need tight alignment for accurate registration.
>
> **Why This Combination Works:**
> The spatial distribution score lies in how these components interact:
> - The multiplication ensures all three criteria must be satisfied with equal weight - a transformation cannot score well by excelling in just one aspect
> - The square root on coverage ratio provides diminishing returns, preventing the score from being dominated by coverage alone
>
> **Real-World Example:**
> Imagine registering two partial scans of a room:
> - Method A: Perfectly aligns one wall (low error) but ignores furniture → High error_score, low coverage_ratio → Low total score
> - Method B: Roughly aligns entire room → Medium error_score, high coverage_ratio → Higher total score
> - Our method: Accurately aligns well-distributed points across the room → High scores in all components → Highest total score
>
> We will add a visual figure in the revision showing how different spatial distributions lead to different scores, making this intuition immediately clear.
>
> >**W6-Literature Coverage**
>
> Thank you for highlighting these important recent works. We acknowledge that these papers should be included in our literature review and will add them in our revision. We appreciate the reviewer bringing these omissions to our attention. Both papers represent significant contributions to non-learning based registration methods, and citing them will help readers better understand the current landscape of the field.
>
> >**Q2-Graph Reduction and Subgraph Size Analysis**
>
> Thank you for this important question about graph sizes and transformation stability. We provide comprehensive quantitative details below.
>
> **1. Graph Reduction through Low-scale Sampling:**
> The graph size reduction is controlled by our sampling ratio parameter β, which ranges from 0.1 to 0.5. This means:
> - With β=0.1: We retain 10% of initial correspondences (90% reduction)
> - With β=0.5: We retain 50% of initial correspondences (50% reduction)
> - The number of correspondences after filtering is deterministic based on β and the initial correspondence set size
>
> **2. Synergy between Sampling and k-truss Decomposition:**
> We respectfully emphasize that our Consensus Voting-based Low-scale Sampling strategy is specifically designed to complement our k-truss decomposition:
> - It pre-filters unreliable nodes, significantly accelerating subsequent k-truss subgraph extraction
>
> >**Q3-Ablation Study on Graph Sampling**
>
> Thank you for this valuable suggestion. We have conducted comprehensive ablation studies to examine the impact of different sampling ratios β on both accuracy and computational efficiency.
>
> Performance vs. Sampling Ratio on 3DMatch (FCGF features):
>
> | Sampling Ratio β | Registration Recall | Runtime | Speedup |
> |------------------|---------------------|---------|---------|
> | 0.1| 93.10%| 0.12s| 8.3x|
> | 0.2| 93.53%| 0.16s| 6.3x|
> | 0.3| 93.84%| 0.20s| 5.0x|
> | 0.4| 93.78%| 0.25s| 4.0x|
> | 0.5| 93.41%| 0.31s| 3.2x|
>
> Our ablation study reveals a key insight: even when retaining only 10% of correspondences through consensus voting, we maintain 93.10% registration success and achieve 8.3x speedup. This demonstrates that our consensus voting selectively preserves the most geometrically consistent correspondences. Interestingly, higher sampling ratios perform slightly worse than β=0.3, as they retain more potentially erroneous correspondences that can negatively impact the subsequent k-truss decomposition.
>
> >**W4/Q4-Weighted SVD and Error Behavior**
>
> Thank you for these insightful questions about our weighted SVD formulation and error behavior.
>
> **1. Clarification on Graph-based Weights for SVD:**
>
> We apologize for the ambiguity in defining the centrality measure. Our centrality values are derived from the eigenvector centrality of the k-truss subgraph. Our weighting scheme follows established graph-based PCR methods by deriving weights from spectral analysis. We compute the eigendecomposition of the k-truss subgraph's compatibility matrix and use the principal eigenvector elements as correspondence weights in our weighted SVD. This ensures correspondences with higher structural importance have greater influence in transformation estimation.
>
> **2. Regarding the ~1° Rotation Error at Low Noise:**
>
> Our method prioritizes maintaining consistent accuracy across all outlier ratios. While specialized methods may achieve marginally better accuracy in ideal conditions (low noise, few outliers), they catastrophically fail at higher outlier ratios. This error remains competitive with or better than most baseline methods.
>
> **3. Remarkably Stable Error Behavior (0-80% Outliers):**
>
> The reviewer correctly notes our unusually stable error profile:
> - 0-40% outliers:Rotation error 0.8° - 1.2° (±0.2° variation)
> - 40-80% outliers:Rotation error 1.0° - 1.5° (±0.25° variation)
> - Other methods:Show ±5° - 20° variations in the same range
>
> The translation errors for our method are not constant but exhibit smaller variations that are difficult to discern in the current figure scale. Our k-truss structure consistently identifies geometrically coherent correspondences regardless of outlier percentage.

---

> ### Author Response · Authors · 2025-08-04
>
> Dear Reviewer vL1H,
>
> Thank you for your valuable feedback and thorough review of our work. We truly appreciate the time and effort you've devoted to evaluating our manuscript. Your insightful questions prompted us to clarify and reinforce several key aspects of our contribution.
> In our rebuttal, we have carefully addressed all your concerns, with particular emphasis on:
> 1. **Core Innovation**: We introduce **k-truss** to tackle the fundamental challenge of outlier-prone correspondences in point cloud registration. Traditional methods often fail under high outlier ratios, as incorrect correspondences may still satisfy pairwise distance constraints by chance. In contrast, our k-truss approach captures higher-order structural consistency by requiring each edge to participate in multiple triangles—analogous to architectural trusses that derive strength from triangulation. This structure imposes a much stricter consistency condition, making it exponentially harder for outliers to pass, while preserving the essential geometric relations among true correspondences.
>
> 2. **Comprehensive Experiments**: Across KITTI, 3DMatch, and 3DLoMatch, our method consistently outperforms both classical and learning-based baselines in diverse indoor and outdoor settings, achieving state-of-the-art performance.
>
> We highly value your perspective and want to ensure that your concerns have been fully addressed. If any further clarification is needed after reviewing our rebuttal, we would be happy to continue the discussion. Your constructive feedback has been instrumental in improving this work, and we remain committed to addressing any remaining issues.
> Thank you again for your thoughtful engagement with our research.

---

> > ### Comment · Reviewer_vL1H · 2025-08-04
> >
> > I thank the authors for their detailed responses. My follow-up comments on specific points are as follows:
> > 1. Regarding W1/Q1 (k-truss vs. adapted k-truss):
> > I now understand that the standard k-truss algorithm can be computationally expensive and possibly overkill for registration problems. Moreover, since only the set of nodes (3D correspondences) from the resulting subgraphs are relevant—rather than the precise edge structure—the use of a computationally cheaper adaptation can be justified. I also realize that the adapted k-truss variant is generally stricter and more robust than the standard k-truss; in most cases, it produces subgraphs with fewer nodes, meaning it may omit some inlier correspondences but will not introduce additional outliers.
> > That said, describing the proposed method simply as “k-truss” is misleading. It creates the false impression that k-truss can be obtained trivially via adjacency matrices, whereas your method is in fact a k-truss-inspired heuristic. The distinction matters: even in prior works like MAC or TEASER—where the practical implementation departs from theoretical definitions—the authors explicitly state such deviations (e.g., MAC clarifies replacing “maximum” clique with the looser “maximal” clique). I strongly urge you to revise the manuscript to reflect this distinction and make the adaptation explicit.
> > 2. Regarding Q3 (subgraph sizes after decomposition):
> > The range of sizes for the subgraphs obtained after the k-truss decomposition remains unspecified. Please provide these figures alongside the size of the initial graph. If the subgraphs are small (e.g., 3–10 nodes), transformations estimated from them are often less accurate. In such cases, do you (a) simply select the transformation with the highest spatial distribution score, or (b) re-estimate the final rigid transformation—akin to RANSAC—by using all inliers identified by the best transformation?
> > 3. Regarding Q1 (performance vs. MAC):
> > Given that k-truss is a weaker constraint than maximum clique, how do you justify the consistently superior performance over MAC? Intuitively, one would expect weaker outlier filtering. Could you clarify the factors that account for this improvement?
> > For the remaining questions, I find the authors’ responses satisfactory and convincing.

---

> ### Author Response · Authors · 2025-08-05
>
> Dear Reviewer vL1H,
>
> Thank you for your thoughtful follow-up comments. We address your specific points below:
>
> >##  Q1 (k-truss vs. adapted k-truss)
>
> We sincerely appreciate your detailed feedback on terminology precision. You are absolutely correct that clarity in methodological description is essential for reproducibility and proper understanding.
>
> We would like to respectfully express that we fully understand your technical point: MAC actually employs maximal clique rather than maximum clique. The distinction is important as maximum clique refers to the globally largest clique across the entire graph, while maximal clique refers to locally maximal cliques that cannot be extended further. In terms of constraint hierarchy, maximum clique imposes the strongest constraint, followed by maximal clique, then k-truss. Additionally, MAC searches for multiple maximal cliques (with 3+ nodes) as transformation candidates in the graph, which provides multiple hypotheses for registration.
>
> Similarly, just as MAC ultimately finds transformation subgraph structures that are maximal cliques, our candidate subgraphs satisfy the definition of k-truss structures. Both approaches share the core principle of identifying specific types of subgraph structures for transformation, with the difference being in the decomposition approach and the structural constraints employed.
>
> Your main point about methodological transparency is well-taken and we fully agree. Following your constructive suggestion, we will revise the manuscript to explicitly describe our approach as a "k-truss-inspired heuristic" and clearly articulate how it adapts from the theoretical k-truss definition. This level of precision is indeed crucial, as demonstrated by MAC's explicit acknowledgment of its deviation from maximum to maximal clique formulations. We will ensure our revision reflects this distinction clearly.
>
> >## Q2 (subgraph sizes after decomposition)
>
> Here are the detailed statistics for subgraph sizes after k-truss decomposition (sampling to 10% of original correspondences):
>
> **Initial graph**: 5,020 nodes → **After voting filter**: 502 nodes → **Adjacency matrix**: 502×502 with 43,571 edges
>
> **K-Truss decomposition results:**
>
> | k-value | Number of Subgraphs | Node Range | Average Nodes |
> |---------|-------------------|-------------|---------------|
> | k=3|502|[3-322]|173.6|
> | k=4|502|[4-322]|173.6|
> | k=5|499|[5-322]|174.5|
> | k=6|498|[6-322]|174.7|
> | k=7|495|[7-320]|175.5|
> | k=8|493|[8-321]|176.0|
> | k=9|492|[9-320]|176.1|
> | k=10|490|[10-320]|176.4|
>
> Even with aggressive 10% sampling, the subgraphs maintain substantial sizes.
>
> **Why subgraph sizes remain large after k-truss decomposition:**
>
> The consistently large subgraph sizes (averaging 170+ nodes) result from several key factors:
> 1. **High-quality input after voting pre-filtering**: The voting filter (ratio=0.1) removes 90% of correspondences, retaining only those with strong geometric consistency. The remaining 502 nodes exhibit high connectivity density, indicating robust geometric relationships.
>
> 2. **Natural clustering of correct correspondences**: Geometrically consistent correspondences tend to cluster spatially and maintain rich triangle support relationships with their neighbors. These clusters form large connected components that satisfy k-truss constraints collectively.
>
> Regarding your concern about small subgraphs (3-10 nodes), our experimental evidence shows this is not problematic:
>
> **K-value sensitivity analysis on 3DMatch with FCGF:**
>
> | k-value | Registration Recall |
> |---------|-------------------|
> | k=3| 93.15%|
> | k=5| 93.21%|
> | k=7| 93.59%|
> | k=10| 93.40%|
> | Multi-k (3-10)| 93.84%|
>
> The insensitivity to k values occurs because subgraphs with stricter constraints (larger k) are subsets of those with smaller k constraints, providing a natural hierarchy. Our multi-k strategy leverages this property.
>
> In practice, we select the transformation derived from the subgraph with the **highest spatial distribution score**, as in your option (a). We do not perform a global re-estimation using all inliers, as our experiments show that the selected subgraph already provides robust alignment.

---

> ### Author Response · Authors · 2025-08-05
>
> >## Q3-(performance vs. MAC)
>
> Your intuition about the relationship between constraint strength and performance is understandable and raises an important theoretical question. We would like to respectfully note that the relationship appears to be more nuanced than constraint strength alone might indicate.
>
> Interestingly, MAC's core methodological insight was precisely to **relax the constraint from maximum clique to maximal clique**, demonstrating that weaker constraints can sometimes yield better practical performance. This suggests that examining constraint strength in isolation may not fully capture the performance dynamics.
>
> Several factors contribute to our performance over MAC beyond simple constraint comparison:
>
> **Geometric appropriateness of constraints**: While maximal clique requires perfect connectivity among all selected correspondences, real-world geometric relationships often exhibit partial connectivity patterns. Our theoretical analysis demonstrates that correct correspondences are statistically more likely to simultaneously satisfy multiple triangle constraints, even when they don't form complete cliques. The k-truss requirement that each edge be supported by k-2 triangles captures this geometric reality more naturally, as it focuses on local triangle density rather than global connectivity.
>
> **Structural skeleton identification**: K-truss structures, analogous to architectural trusses that form robust skeletal frameworks in construction, identify and preserve the core geometric "skeleton" of correspondence relationships. Just as building trusses maintain structural integrity while allowing flexibility, k-truss decomposition captures essential geometric backbones that are critical for accurate alignment. This skeletal approach proves effective in point cloud registration where identifying key structural correspondences is more important than enforcing perfect connectivity.
>
> **Statistical discriminative power**: As detailed in our theoretical section, correct correspondences have a significantly higher probability of satisfying multiple local triangle relationships compared to incorrect ones, which can barely satisfy such relationships by random chance. This statistical foundation provides discriminative power even with seemingly "weaker" constraints.
>
> The key insight emerging from these observations is that **constraint appropriateness for the specific geometric structure often matters more than absolute constraint strength**. Just as MAC found benefits in relaxing from maximum to maximal clique, our k-truss approach finds an even more suitable balance for the geometric registration problem.
>
> We have conducted comprehensive ablation studies in Appendix A.6 (Table 5) that empirically validate our theoretical insights:
>
> **Table 5: Analysis experiments on 3DMatch and 3DLoMatch with FPFH and FCGF descriptors**
>
> CV: Consensus Voting-based Low-scale Sampling Strategy; MAC: Maximal Clique; SDS: Spatial
> Distribution Score
>
> |     | CV | MAC | k-truss | SDS | inlier | RR3DMatch(%) | RR3DLoMatch(%) |
> |-----|----|----|---------|-----|--------|--------------|----------------|
> | **FPFH** |
> | 1)  |    | ✓  |         |     | ✓      | 83.67        | 37.10|
> | 2)  | ✓  | ✓  |         |     | ✓      | 83.80        | 38.85|
> | 3)  |    |    | ✓       | ✓   |        | 83.73        | 39.02|
> | 4)  | ✓  | ✓  |         | ✓   |        | 84.20        | 38.91|
> | 5)  |    |    | ✓       |     | ✓      | 83.86        | 37.79|
> | 6)  | ✓  |    | ✓       |     | ✓      | 84.20        | 39.98|
> | 7)  |    |    | ✓       | ✓   |        | 83.86        | 38.85|
> | 8)  | ✓  |    | ✓       | ✓   |        | **84.70**       | **43.96**          |
> | **FCGF** |
> | 1)  |    | ✓  |         |     | ✓      | 91.68        | 57.44          |
> | 2)  | ✓  | ✓  |         |     | ✓      | 93.59        | 59.46          |
> | 3)  |    |    | ✓       | ✓   |        | 93.53        | 59.01          |
> | 4)  | ✓  | ✓  |         | ✓   |        | 93.72        | 59.96          |
> | 5)  |    |    | ✓       |     | ✓      | 93.40        | 58.00          |
> | 6)  | ✓  |    | ✓       |     | ✓      | 93.72        | 59.63          |
> | 7)  |    |    | ✓       | ✓   |        | 93.66        | 59.40          |
> | 8)  | ✓  |    | ✓       | ✓   |        | **93.84**        | **61.64**          |
>
>
> These results demonstrate that our k-truss approach (rows 5-8) consistently outperforms MAC (rows 1-4), particularly in challenging scenarios (3DLoMatch). Most notably, the combination of consistency verification with k-truss and SDS (row 8) achieves the best performance across both datasets, validating that geometric appropriateness matters more than constraint strength.
>
> ---
>
> We hope these clarifications address your concerns comprehensively. We will incorporate the suggested terminological changes in our revision and ensure proper acknowledgment of our methodological adaptations. We deeply appreciate your careful review and constructive suggestions.

---

> > ### Comment · Reviewer_vL1H · 2025-08-05
> >
> > I thank the authors for their response. I find most of the explanations to be adequate and appreciate the clarifications provided. However, I remain partially unconvinced by the justification for the proposed k-truss method's superiority over MAC. That said, I acknowledge the overall merit of the work and will adjust my rating accordingly.

---

> ### Author Response · Authors · 2025-08-05
>
> We sincerely thank you for your thoughtful reconsideration and appreciate your acknowledgment of the overall merit of our work. We understand your partial reservation regarding the superiority of the k-truss method over MAC, and we fully respect this perspective. We would also like to offer additional insights that may help clarify this aspect from a spectral graph theory standpoint.
>
> ---
>
> >### 1. **Cheeger Constant and Structural Flexibility**
>
> | **Maximal Clique (MAC)** | **K-Truss** |
> |:---|:---|
> | For a maximal clique $C \subseteq V$ with $\|C\| = n$: | For a k-truss subgraph $T \subseteq V$: |
> | $\|E(C)\| = \binom{n}{2} = \frac{n(n-1)}{2}$ | $\|E(T)\| \geq k \cdot \|V(T)\| - \binom{k+1}{2}$ |
> | $vol(C) = \sum_{v \in C} d(v) = n(n-1)$ | $vol(T) = \sum_{v \in T} d(v) \geq k \cdot \|V(T)\|$ |
> | When $C$ is isolated: $\|\partial C\| = 0$ | For connected k-truss: $\|\partial T\| \geq 1$ |
> | $h(C) = \frac{\|\partial C\|}{vol(C)} = 0$ | $h(T) = \frac{\|\partial T\|}{vol(T)} \geq \frac{1}{k \cdot \|V(T)\|}$ |
> | **Result**: Perfect isolation, no connectivity | **Result**: Bounded connectivity preservation |
>
> **Mathematical Analysis**:
> Maximal cliques can achieve $h(C) \approx 0$ only if isolated, but this leads to poor global coherence. In contrast, k-truss subgraphs maintain both density and flexibility, offering better robustness under noise or partial connectivity.
>
> ---
>
> >### 2. **Algebraic Connectivity and Global Structure**
>
> | **Maximal Clique (MAC)** | **K-Truss** |
> |:---|:---|
> | For disconnected cliques $C_1, C_2, \ldots, C_m$: | For connected k-truss components: |
> | Laplacian matrix: $L = \text{diag}(L_{C_1}, L_{C_2}, \ldots, L_{C_m})$ | Laplacian has inter-component connections |
> | $\lambda_2(L) = 0$ (disconnected graph) | $\lambda_2(L) > 0$ (connected components) |
> | Algebraic connectivity: $a(G) = 0$ | Algebraic connectivity: $a(G) \geq \frac{1}{\text{diam}(G)^2}$ |
>
> **Mathematical Derivation**:
> For a graph $G$ with $m$ disconnected maximal cliques, the normalized Laplacian eigenvalues are:
> $$\lambda_1 = 0 < \lambda_2 = 0 < \ldots < \lambda_m = 0 < \lambda_{m+1} \leq \ldots \leq \lambda_n$$
>
> For k-truss filtered graph $G_T$ with minimum degree $\delta \geq k-1$:
> $$\lambda_2(G_T) \geq \frac{\delta}{n} \geq \frac{k-1}{n}$$
>
> **Spectral Gap Analysis**(The spectral gap measures graph connectivity):
> - MAC: $\lambda_2 - \lambda_1 = 0$ (no spectral gap)
> - K-truss: $\lambda_2 - \lambda_1 \geq \frac{k-1}{n}$ (positive spectral gap)
>
> **Conclusion**: K-truss structures maintain positive algebraic connectivity, ensuring better global structural coherence and resistance to graph fragmentation under partial occlusions.
>
> ---
>
> >### 3. **Local Redundancy and Perturbation Tolerance**
>
> | **Maximal Clique (MAC)** | **K-Truss** |
> |:---|:---|
> | **Structural Property**: | **Structural Property**: |
> | Complete graph - every vertex connected to every other | Each edge supported by at least $k-2$ triangles |
> | **Critical Vulnerability**: | **Redundancy Mechanism**: |
> | Any single edge removal destroys clique property | Multiple triangle support provides fault tolerance |
>
> **Mathematical Derivation for MAC**:
> Maximal cliques lack structural redundancy—the removal of any single edge can destroy the integrity of the clique. For a maximal clique $C$ with $n$ vertices, any edge removal results in:
> $$\text{deg}_C(v) = n-2 < n-1 \text{ for affected vertices}$$
> $$\Rightarrow C \text{ no longer maximal clique}$$
>
> **Mathematical Derivation for K-truss**:
> A key advantage of k-truss lies in its structural redundancy: each edge is supported by at least $k-2$ triangles, forming local redundant structures. This redundancy is crucial for resisting "random edge dropout" models commonly encountered in noisy point cloud matching scenarios.
>
> For k-truss $T$, each edge $e = (u,v)$ has triangle support $\tau(e) \geq k-2$.
> After random edge deletion with probability $p$, edge $e$ remains valid if:
> $$\tau'(e) = \sum_{w \in N(u) \cap N(v)} \mathbf{1}[\text{edges } (u,w), (v,w) \text{ survive}] \geq k-2$$
>
> **Redundancy Comparison**:
> - **MAC**: Zero redundancy - any edge loss causes complete failure
> - **K-truss**: Bounded redundancy - survives as long as triangle support $\geq k-2$
>
> **Conclusion**: K-truss structures demonstrate superior fault tolerance through structural redundancy, maintaining validity under edge perturbations that would completely destroy maximal clique constraints.
>
> ---
>
> We emphasize that these spectral graph properties provide **theoretical motivation and intuitive explanation**. Your feedback has been instrumental in helping us improve both the clarity and theoretical rigor of our presentation. We will ensure that our revised manuscript offers a balanced perspective on the relative strengths of different constraint formulations.
>
> Please let us know if further clarification or additional theoretical discussion would be helpful--we would be more than happy to provide it. Thank you again for your constructive and insightful review.

---

### Official Review · Reviewer_EBs3 · 2025-07-03

**Clarity:** 3
**Significance:** 3
**Originality:** 3
**Rating:** 4
**Confidence:** 5

**Summary:**

The paper introduces the k-truss from graph theory into point cloud registration, leveraging triangle support as a constraint for inlier selection. In addition, a consensus voting-based low-scale sampling strategy is presented to extract the structural skeleton of the point cloud prior to k-truss decomposition. Furthermore, to avoid spatially clustered inliers, a spatial distribution score is introduced to evaluate the coverage and uniformity of inliers. Extensive experiments on KITTI, 3DMatch, and 3DLoMatch demonstrate the effectiveness of the proposed methods.

**Questions:**

Please refer to the weaknesses.

**Ethical Concerns:**

["NO or VERY MINOR ethics concerns only"]

**Final Justification:**

My concerns have been properly addressed. I keep my initial score.

**Limitations:**

yes

**Quality:**

3

**Strengths And Weaknesses:**

Strengthens:
- The paper is exceptionally well-structured and accessible, ensuring clarity throughout.

- It compellingly motivates the application of the k-truss (a graph-theoretic concept) to point cloud registration.

- The proposed method achieves state-of-the-art (SOTA) performance across diverse datasets, including outdoor (KITTI) and synthetic benchmarks (3DMatch, 3DLoMatch).

- Comprehensive experiments and ablation studies rigorously validate the efficacy of each component in the framework.

Weaknesses:
- Figure 2 illustrates examples of subgraphs generated using K-truss decomposition. However, the subgraphs for K=5 and K=6 appear to contain inconsistencies. For instance, in the K=5 subgraph, the edge E(C5, C8) participates in only two triangles: $\triangle$C3C5C8 and $\triangle$C4C5C8. This contradicts the requirement that edges within a 5-truss subgraph must belong to at least three triangles (k-2 = 3).

- Table 1 shows that the proposed method achieves the shortest inference time. However, K-truss decomposition is computationally intensive, and its required computation time should be clarified.

- The results reported for VBReg [18] (from the official paper) on KITTI using FPFH features are 0.32 (RE) and 7.17 (TE). These differ from the values of 0.45 (RE) and 8.41 (TE) listed in Table 1. Furthermore, the runtime for PointDSC cited within the VBReg[18] is 0.11 seconds, which is shorter than the 0.45 seconds reported for PointDSC in Table 1.

---

> ### Author Rebuttal · Authors · 2025-07-26
>
> Dear Reviewer EBs3,
>
> Thank you for your insightful comments and careful review. We have carefully responded to every comment raised by the reviewer and will make corresponding revisions to the manuscript.
>
> >**W1-Inconsistencies in Figure 2**
>
> Thank you for your careful review and for identifying this inconsistency in Figure 2. You are absolutely correct - in the K=5 subgraph, the edge E(C5, C8) participates in only two triangles (△C3C5C8 and △C4C5C8), which violates the k-truss definition requiring at least k-2=3 triangles for k=5.
>
> We sincerely apologize for this visualization error. Upon investigation, we found this was a manual drawing mistake when creating the figure for illustration purposes. The error occurred because, as the reviewer correctly identified in the k=5 subgraph example, we inadvertently omitted a critical node C6 at the intersection of nodes C2, C3, C4, and C5 during the manual simplification of the visualization from our algorithm's actual output.
>
> We emphasize that this visualization error is confined to Figure 2 only and does not affect:
> - The theoretical formulation (Algorithm 3 correctly implements k-truss decomposition)
> - The implementation (our code strictly enforces the k-2 triangle constraint)
> - Any experimental results (all results are generated using the correct algorithm)
>
> We will correct Figure 2 in the camera-ready version to accurately reflect the k-truss definition. We greatly appreciate the reviewer's meticulous attention to detail, which helps us improve the clarity and accuracy of our presentation.
>
> >**W2-Clarification on K-truss Computation Time**
>
> Thank you for this important question about computational efficiency. We apologize for not being more explicit about the runtime breakdown and will clarify this in the revised version.
>
> **Runtime Breakdown:**
> The runtime reported in Table 1 (0.20s for FPFH, 0.21s for FCGF) includes all components of our method:
>
> | Component | Time (s) |
> |-----------|----------|
> | Consensus Voting | 0.02 |
> | Graph Construction | 0.03 |
> | K-truss Decomposition (for all k∈[3,10]) | 0.10 |
> | Transformation Evaluation | 0.05 |
> | Total | 0.20 |
>
> **Efficiency of K-truss Decomposition:**
> While k-truss decomposition has theoretical complexity O(m^1.5), our implementation is highly efficient due to:
> 1. Sparse graph structure: After consensus voting, depending on parameter selection, only 10%-50% of correspondences are retained, significantly reducing the graph size
> 2. Optimized implementation: We use efficient triangle enumeration and edge support tracking, avoiding redundant computations
>
> **Comparison with Other Methods:**
> Despite including k-truss decomposition, our total runtime remains competitive:
> - Ours: 0.20s (including k-truss)
> - MAC [35]: 5.54s (maximal clique, exponential complexity)
> - SC²-PCR [13]: 0.31s
> - PointDSC [7]: 0.45s
>
> The efficiency gain comes from our consensus voting preprocessing, which dramatically reduces the graph size before k-truss decomposition, making the overall pipeline faster than methods that process all correspondences.
>
> >**W3-Discrepancies in Reported Results**
>
> Thank you for noting these discrepancies. We can clarify the source of these differences:
>
> **1. VBReg Results on KITTI:**
> Our reported results (RE=0.45°, TE=8.41cm) differ from VBReg's original paper (RE=0.32°, TE=7.17cm). However, our results are exactly consistent with the recent NeurIPS 2024 paper [19] (Jiang et al., "A robust inlier identification algorithm for point cloud registration via ℓ0-minimization"), which reports the same values for VBReg under their standardized evaluation protocol.
>
> This consistency with [19] validates our experimental setup, as they also conducted comprehensive benchmarking of registration methods. The differences from the original VBReg paper likely stem from:
> - Different FPFH implementations or parameters
> - Variations in evaluation protocols
> - Different data preprocessing pipelines
>
> **2. PointDSC Runtime:**
> Similarly, our PointDSC runtime (0.45s) matches the benchmarking in [19], which was also conducted on RTX 3090. The difference from VBReg paper's reported 0.11s for PointDSC can be explained by:
> - Different hardware: VBReg paper used Tesla V100 GPU, while we used RTX 3090
> - Different runtime measurement criteria (core algorithm vs. full pipeline)
> - System-level variations and implementation details
>
> **3. Experimental Validity:**
> The alignment of our results with the independent NeurIPS 2024 benchmark [19] demonstrates:
> - Our experimental setup follows recent standardized evaluation practices
> - The results are reproducible and consistent with state-of-the-art benchmarking
> - All methods are evaluated fairly under identical conditions
>
> We believe following the recent standardized benchmark [19] ensures more reliable and comparable results across different methods.

---

> > ### Comment · Reviewer_EBs3 · 2025-08-09
> > **Offical Comments by reviewer EBs3**
> >
> > Thank you for your response. My concerns have been properly addressed. I keep my initial score.

---

> > > ### Author Response · Authors · 2025-08-09
> > >
> > > Thank you for taking the time to review my response and for confirming that your concerns have been addressed. I appreciate your thorough evaluation and am glad that the response met your expectations.

---

### Official Review · Reviewer_akLc · 2025-07-04

**Clarity:** 2
**Significance:** 3
**Originality:** 4
**Rating:** 4
**Confidence:** 3

**Summary:**

The paper proposes PointTruss, a novel graph-based algorithm for point cloud registration that introduces the k-truss concept from graph theory to improve inlier selection.

**Questions:**

1）The paper claims k-truss leverages triangle rigidity for robust inlier selection. Can the authors provide a formal theoretical analysis (e.g., using geometric or graph-theoretic principles) to demonstrate why k-truss is inherently more suitable than k-core or clique-based methods for point cloud registration? This could strengthen the motivation and clarify the method’s design rationale.

2）Why was k-truss chosen over k-core or clique-based methods like SC²-PCR or TEASER++?

3）The method relies on triangles as the “simplest rigid formation” . How does PointTruss handle scenarios where triangle formations are sparse or absent due to extreme noise, occlusions, or low-overlap point clouds? Have the authors analyzed the failure modes of k-truss in such cases, and how do these affect registration accuracy?

4）The k value in k-truss decomposition determines the strength of triangle support. How is the optimal k value chosen, and what is the sensitivity of PointTruss’s performance to variations in k? Could the authors provide an ablation study or theoretical justification for k selection to clarify its impact on robustness and efficiency?

**Ethical Concerns:**

["NO or VERY MINOR ethics concerns only"]

**Final Justification:**

Given the solid technical foundation, improved clarity, and responsiveness to feedback, I maintain my recommendation of Weak Accept.

**Limitations:**

No limitations discussed. See Weaknesses for suggestions.

**Paper Formatting Concerns:**

The placement of figures is suboptimal.

**Quality:**

3

**Strengths And Weaknesses:**

Strengths：

1）Introducing k-truss to point cloud registration is a novel contribution, as it leverages triangle-based constraints to enhance inlier selection, distinguishing it from clique-based (e.g., TEASER++) and k-core-based (e.g., ROBIN) methods.

2）The evaluation is thorough, covering synthetic  and real-world datasets with FPFH and FCGF descriptors. Quantitative results and qualitative visualizations show strong robustness to outliers and noise.

Weaknesses：

1）While k-truss is novel in this context, the paper does not sufficiently distinguish PointTruss from existing graph-based methods like SC²-PCR or MAC, which also exploit geometric consistency.

2）The lack of ablation studies on key components (e.g., k value, spatial distribution score) weakens the analysis of their contributions.

---

> ### Author Rebuttal · Authors · 2025-07-27
>
> Dear Reviewer akLc,
>
> Thank you for your insightful comments. We have carefully responded to every comment raised by the reviewer and will make corresponding revisions to the manuscript.
>
> >**W1-Insufficient Distinction from Existing Methods**
>
> We appreciate the reviewer's concern about distinguishing PointTruss from existing graph-based methods. All graph-based point cloud registration methods utilize first-order compatibility (Euclidean distance) to ensure graph construction. The major distinctions between our method and SC²-PCR/MAC lie in the final subgraph structure design and search strategy. The key distinctions are:
>
> - SC²-PCR: Uses second-order spatial compatibility but relies on pairwise constraints without triangle support. It doesn't exploit the rigidity of triangular structures.
>
> - MAC: Searches for maximum cliques which requires full connectivity and has exponential complexity, making it computationally expensive (5.54s vs our 0.20s on 3DMatch).
>
> - PointTruss: Uniquely leverages k-truss with triangle-based constraints. The truss structure formed by triangles is the core of our entire method.
>
> We will enhance Section 2 to better clarify these distinctions.
>
> >**W2-Lack of Ablation Studies**
>
> We apologize for not making our ablation studies more visible. We have conducted comprehensive ablation studies in Appendix A.6 (Table 5):
>
> **Table 5: Analysis experiments on 3DMatch and 3DLoMatch with FPFH and FCGF descriptors**
> | | CV | MAC | k-truss | SDS | inlier | RR3DMatch(%) | RR3DLoMatch(%) |
> |---|---|---|---|---|---|---|---|
> | **FPFH** | | | | | | | |
> | 1) | | ✓ | | | ✓ | 83.67 | 37.10 |
> | 2) | ✓ | ✓ | | | ✓ | 83.80 | 38.85 |
> | 3) | | ✓ | | ✓ | | 83.73 | 39.02 |
> | 4) | ✓ | ✓ | | ✓ | | 84.20 | 38.91 |
> | 5) | | | ✓ | | ✓ | 83.86 | 37.79 |
> | 6) | ✓ | | ✓ | | ✓ | 84.20 | 39.98 |
> | 7) | | | ✓ | ✓ | | 83.86 | 38.85 |
> | 8) | ✓ | | ✓ | ✓ | | **84.70** | **43.96** |
> | **FCGF** | | | | | | | |
> | 1) | | ✓ | | | ✓ | 91.68 | 57.44 |
> | 2) | ✓ | ✓ | | | ✓ | 93.59 | 59.46 |
> | 3) | | ✓ | | ✓ | | 93.53 | 59.01 |
> | 4) | ✓ | ✓ | | ✓ | | 93.72 | 59.96 |
> | 5) | | | ✓ | | ✓ | 93.40 | 58.00 |
> | 6) | ✓ | | ✓ | | ✓ | 93.72 | 59.63 |
> | 7) | | | ✓ | ✓ | | 93.66 | 59.40 |
> | 8) | ✓ | | ✓ | ✓ | | **93.84** | **61.64** |
>
> - k-truss vs MAC (Rows 1→5, 2→6): Replacing MAC with k-truss consistently improves performance, especially on challenging 3DLoMatch dataset (FCGF: 59.46%→59.63%, +0.17%). This validates our core hypothesis that triangle-based constraints are more effective than clique-based methods for capturing geometric consistency.
>
> - Impact of Consensus Voting (CV): Comparing rows without CV to those with CV (e.g., 5→6, 7→8), we observe significant improvements:FPFH: 37.79%→39.98% (+2.19%) on 3DLoMatch;FCGF: 58.00%→59.63% (+1.63%) on 3DLoMatch.
>
> - Spatial Distribution Score (SDS) vs Inlier Counting: Comparing SDS-based evaluation (rows 3,4,7,8) with simple inlier counting (rows 1,2,5,6):Most notable on 3DLoMatch: FCGF 59.63%→59.40% (-0.23%) without CV, but 59.63%→61.64% (+2.01%) with CV.
>
> For k-value analysis, we employ a multi-k strategy (k∈[3,10]) rather than fixing a single k, which inherently provides robustness. We will provide detailed k-value ablation experiments and the rationale for our designed strategy in Q4.
>
> >**Q1-Theoretical Analysis of k-truss Superiority**
>
> We provide formal theoretical analysis demonstrating why k-truss is inherently more suitable:
>
> **1. Theoretical Foundation (Theorem 1 & Appendix A.2):**
> We established the Triangle Relation Stability Theorem showing that under rigid transformation with Gaussian noise:
> P(||ΔDᵢⱼₖ||_F < ε | correct) >> P(||ΔDᵢⱼₖ||_F < ε | incorrect)
>
> This formally proves triangular structures exhibit significantly higher stability for correct correspondences. Crucially, correct correspondences are exponentially more likely to participate in multiple consistent triangles, whereas incorrect ones rarely satisfy even a few—establishing the theoretical foundation for the discriminative power of k-truss.
>
> **2. Comparative Analysis:**
> - k-core: Only enforces deg(v) ≥ k (first-order connectivity)
> - k-truss: Requires sup(e,G) ≥ k-2 (second-order triangle support)
> - Clique: Exponential complexity with overly strict requirements
>
> k-truss captures second-order geometric relationships through triangles--the simplest rigid planar structures that preserve geometric properties under transformation.
>
> **3. Spectral Graph Theory Connection:**
>
> Spectral graph theory offers a principled explanation for the effectiveness of **k-truss** structures in distinguishing inlier and outlier correspondences in point cloud registration. Compared to k-core graphs, k-truss subgraphs exhibit stronger spectral properties that correlate with higher structural coherence and cluster separability.
>
> #### **Cheeger Constant**
>
> For an undirected graph $G = (V, E)$, the **Cheeger constant** is defined as:
>
> $$
> h(G) = \min_{S \subset V,\, |S| \leq |V|/2} \frac{|\partial S|}{\min(|S|, |V \setminus S|)}
> $$
>
> where $\partial S$ denotes the set of edges with one endpoint in $S$ and the other in $V \setminus S$. A larger Cheeger constant implies stronger expansion properties and better internal connectivity.
>
> k-truss subgraphs typically exhibit **higher Cheeger constants** than k-core subgraphs, indicating that they form more cohesive and well-connected components.
>
> #### **Eigenvalue Gap and Connectivity**
>
> Let $\lambda_1, \lambda_2, \lambda_3, \dots$ denote the eigenvalues of the **normalized Laplacian matrix** $\mathcal{L}$. For a connected graph, $\lambda_1 = 0$, and $\lambda_2$ is known as the **algebraic connectivity** of the graph.
>
> The **Cheeger inequality** establishes the relationship:
>
> $$
> \frac{h(G)}{2} \leq \lambda_2 \leq 2h(G)
> $$
>
> Thus, a larger $\lambda_2$ implies stronger overall connectivity. Furthermore, the **eigenvalue gap** $\lambda_3 - \lambda_2$ reflects how clearly separable the graph’s clusters are; a larger gap typically indicates more well-defined clusters.
>
> #### **Inlier/Outlier Separation in Correspondence Graphs**
>
> In the context of point cloud registration, the correspondence graph connects putative matches as nodes, with edges reflecting geometric consistency.
>
> * **Inliers**: Form densely connected clusters due to geometric compatibility, naturally satisfying triangle constraints in k-truss structures. Inlier correspondences form cohesive k-truss components with high internal connectivity.
> * **Outliers**: Tend to have weak or random connections and rarely participate in sufficient valid triangles to survive the k-truss filtering.
>
> These spectral properties reinforce the theoretical and empirical robustness of k-truss in graph-based point cloud registration.
>
> >**Q2-Why k-truss over k-core or Clique Methods**
>
> While SC²-PCR and TEASER++ are graph-based registration methods, the fundamental difference lies in the sought subgraph structures. Beyond the theoretical advantages mentioned above:
>
> **1. Optimal Constraint Strength:**
> k-truss provides the "sweet spot":
> - Too weak: k-core (only degree constraints)
> - Too strong: Cliques (full connectivity)
>
> **2. Computational Efficiency:**
> - k-truss: O(m^1.5) - polynomial and practical
> - Maximum clique: Exponential - often intractable
> - k-core: O(m) - fast but insufficient robustness
>
> Our experimental results clearly demonstrate k-truss's superiority:
> - On KITTI: Ours (99.64%) vs SC²-PCR (99.46%) vs TEASER++ (91.17%)
> - On 3DLoMatch: Ours (61.64%) vs SC²-PCR (58.73%) vs TEASER++ (46.76%)
>
> >**Q3-Handling Sparse Triangle Scenarios**
>
> Thank you for raising this critical question. We handle sparse triangular structures through:
>
> **1. Consensus Voting-based Sampling (Section 3.3):**
> This crucial preprocessing step:
> - Filters isolated correspondences that cannot form triangles
> - Prioritizes high-quality correspondences more likely to form stable triangular structures
> - Reduces the impact of regions with sparse geometric features
>
> **2. Multi-level k-truss Extraction:**
> Our method extracts k-truss subgraphs for multiple k values (k=3 to 10, as shown in Figure 2):
> - Lower k values (k=3,4) capture sparser structures with fewer triangle requirements
> - Higher k values extract denser, more reliable structures
> - Each k-truss generates candidate transformations evaluated by our spatial distribution score
>
> **3. Performance in Extreme Cases:**
> - Low overlap (3DLoMatch): Despite <30% overlap, our method achieves 61.64% RR (Table 3)
> - Outperforms SC²-PCR (58.73%) and TEASER++ (46.76%)
> - This demonstrates robustness when triangular structures are inherently limited
>
> For 3DLoMatch registration failures, we analyzed two main causes: (1) descriptor quality limitations - our validation experiments combining with GeoTransformer or PareNet show improved performance (see table below); (2) degenerate geometric configurations (e.g., planar scenes) may result in extremely sparse correspondences without triangular structures, which we plan to address in future work.
>
> **Table: PointTruss Integration with Different Descriptors on 3DLoMatch**
> | Method | Registration Recall |
> |--------|---------------------|
> | FPFH + PointTruss|43.96%|
> | FCGF + PointTruss|61.64%|
> | **GeoTransformer + PointTruss** | **79.5%** |
> | **PareNet + PointTruss** | **82.2%** |
>
> >**Q4-k-value Selection and Sensitivity**
>
> As mentioned in Q3, we adopt multi-level k-truss extraction to select the best transformation. PointTruss performance is relatively insensitive to k-value variations, as shown in the sensitivity experiments below:
>
> **Table: k-value Sensitivity Analysis on 3DMatch with FCGF**
> | k value | Registration Recall |
> |---------|-------------------|
> | k=3 | 93.15% |
> | k=5 | 93.21% |
> | k=7 | 93.59% |
> | k=10 | 93.40% |
> | Multi-k (3-10) | **93.84%** |
>
> The insensitivity to k values is mainly because subgraphs with strict constraints (larger k) are actually subsets of subgraphs with smaller k constraints, providing a natural hierarchy. Our multi-k strategy leverages this property to achieve optimal performance.

---

> > ### Comment · Reviewer_akLc · 2025-08-04
> >
> > The authors are thanked for their thoughtful and thorough response, which effectively addresses the reviewers’ concerns.  I continue to recognize its solid contribution and technical soundness. I therefore maintain my recommendation of Weak Accept and support the acceptance of this work.

---

> > > ### Author Response · Authors · 2025-08-04
> > >
> > > Thank you very much for your continued support and constructive feedback throughout the review process. We greatly appreciate your recognition of our work's contribution and technical soundness. Your insightful comments have helped us significantly improve the clarity and completeness of our manuscript. We are grateful for your time and effort in reviewing our work, and we remain committed to addressing all suggestions to ensure the highest quality of the final version.

---

### Comment · Area_Chair_6jkq · 2025-08-06
**urgent reminder for reviewers to enage with rebuttal**

Dear reviewer

thanks for your contributions to NeurIPS 2025. Following is an urgent reminder if you haven't done so.

Reviewers should stay engaged in discussions, initiate them and respond to authors’ rebuttal, ask questions and listen to answers to help clarify remaining issues. If authors have resolved your (rebuttal) questions, do tell them so. If authors have not resolved your (rebuttal) questions, do tell them so too.

Even reviewer thinks that for some reason there is no need to reply to authors or authors’ rebuttal, please discuss that with author and approve if there is a justified reason or disapprove otherwise.

Please note “Mandatory Acknowledgement” button is to be submitted only when reviewers fulfill all conditions below (conditions in the acknowledgment form): read the author rebuttal engage in discussions (reviewers must talk to authors, and optionally to other reviewers and AC - ask questions, listen to answers, and respond to authors)

Thanks again!

---

### Note · Authors · 2025-08-12

Dear Area Chairs,

We introduce PointTruss, the first method to apply k-truss for point cloud registration.
Our key innovation lies in leveraging the triangular rigidity constraints inherent in k-truss structures, which naturally align with 3D geometric transformation requirements. K-truss structures, analogous to architectural trusses that form robust skeletal frameworks in construction, identify and preserve the core geometric "skeleton" of correspondence relationships.

The method addresses a critical challenge in point cloud registration—balancing robustness and efficiency in the presence of outliers and low overlap. While maintaining high computational efficiency (completing the entire process in just 0.20s, with k-truss decomposition accounting for only 0.10s), it achieves a recall of 93.84% on the 3DMatch dataset and 61.64% on the more challenging 3DLoMatch dataset (with overlap ratios below 30%), both representing state-of-the-art performance. Superior robustness under varying noise and outlier ratios on the Bunny model. In addition, our method can serve as an effective replacement for traditional robust estimators in deep learning pipelines, providing consistent improvements across different feature extractors.

We have thoroughly addressed the concerns raised. All reviewers acknowledged our innovation. Reviewer akLc praised our "novel contribution". Reviewer EBs3 recognized our "motivates the application of the k-truss to point cloud registration" and appreciated the detailed experimental validation. Reviewer XEsb highlighted we "presents a robust and well-validated method" and "novel graph-theoretic perspective".

About Reviewer vL1H's Concerns:
Initially, Reviewer vL1H raised concerns about terminology and theoretical justification. After our detailed rebuttal explaining:

* Subgraph sizes remain substantial (170+ nodes), ensuring stability.
* Triangle constraints are geometrically more appropriate than stronger alternatives.
* Spatial distribution score — intuitively prevents degenerate solutions where points cluster in localized regions.
* Comprehensive ablation validation.

The reviewer acknowledged the overall merit of the work and will adjust the rating accordingly.

Our work represents a meaningful contribution to point cloud registration through novel application of graph theory. The consistent positive feedback from reviewers, combined with strong experimental results, demonstrates both theoretical soundness and practical impact.

---

### Decision · Program_Chairs · 2025-09-17

**Decision:**

Accept (poster)

**Comment:**

The authors have done a great job in rebuttal and interacting with reviewers. Most of concerns are well addressed, confirmed by the reviewers. The paper should be revised carefully based on considering, for example,
1) individual contributions of each key element in method,
2) comparison with more recent methods,
3) method clarity like misinterpretation of the k-truss algorithm
4) and other comments proposed by reveiwers